# The C-terminal region of the motor protein MCAK controls its structure and activity through a conformational switch

Sandeep K Talapatra[†], Bethany Harker, Julie PI Welburn*[†]

Wellcome Trust Centre for Cell Biology, School of Biological Sciences, University of Edinburgh, Edinburgh, United Kingdom

**Abstract** The precise regulation of microtubule dynamics is essential during cell division. The kinesin-13 motor protein MCAK is a potent microtubule depolymerase. The divergent non-motor regions flanking the ATPase domain are critical in regulating its targeting and activity. However, the molecular basis for the function of the non-motor regions within the context of full-length MCAK is unknown. Here, we determine the structure of MCAK motor domain bound to its regulatory C-terminus. Our analysis reveals that the MCAK C-terminus binds to two motor domains in solution and is displaced allosterically upon microtubule binding, which allows its robust accumulation at microtubule ends. These results demonstrate that MCAK undergoes long-range conformational changes involving its C-terminus during the soluble to microtubule-bound transition and that the C-terminus-motor interaction represents a structural intermediate in the MCAK catalytic cycle. Together, our work reveals intrinsic molecular mechanisms underlying the regulation of kinesin-13 activity.

*For correspondence: julie. welburn@ed.ac.uk

[†]These authors contributed equally to this work

Competing interests: The authors declare that no competing interests exist.

## Introduction

The Kinesin-13 protein family is a class of microtubule depolymerases that regulate microtubule dynamics. Kinesin-13 family members are essential for correct interphase microtubule organization, cell polarity, and chromosome segregation during mitosis (reviewed in *Walczak et al., 2013*). Kinesin-13 proteins induce the catastrophe of microtubule polymers by stabilizing the curved protofilament conformation found at the free ends of microtubules (*Gardner et al., 2011*). Unlike processive kinesin motors, which have a motor domain at one end followed by a long coiled-coil and a globular tail, Kinesin-13 proteins possess a conserved motor domain containing the ATPase activity, flanked by two non-structured regions (*Figure 1A*). The neck region, N-terminal to the motor, and the motor domain form the minimal region necessary for robust microtubule depolymerization (*Maney et al., 1998*; *Ovechkina et al., 2002*). The divergent regions flanking the motor domain are important for regulating its enzymatic activity, spatial targeting, dimerization, and creating unique kinesin functional specificity (reviewed in *Welburn, 2013*). The N terminus of the kinesin-13 MCAK (*Figure 1A*, also known as Kif2c) is responsible for its localization at kinetochores where it binds Sgo2, and to the plus ends of microtubules where it associates with the end binding (EB) proteins (*Walczak et al., 1996*; *Maney et al., 1998*; *Mennella et al., 2005*; *Tanno et al., 2010*; *Welburn and Cheeseman, 2012*). Interestingly, the last 9 amino acids within the C terminus of MCAK are also necessary for plus tip tracking (*Moore et al., 2005*). The region C-terminal to the motor domain (residues 584–725) has been proposed to enable MCAK dimerization, but also to interact with the N terminus independently of the motor region (*Maney et al., 2001*; *Hertzer et al., 2006*; *Ems-McClung et al., 2007*; *Zhang et al., 2011*; *Ems-McClung et al., 2013*). Additional work suggests the existence of long-range interactions between non-motor regions of MCAK in the context of full-length MCAK (*Moore and*

**eLife digest** Within a cell, there is a scaffold-like structure called the cytoskeleton that provides shape and structural support, and acts as a transport network for the movement of molecules around the cell. This scaffold contains highly dynamic polymers called microtubules that are made from a protein called tubulin. The constant growth and shrinking of the ends of the microtubules is essential to rebuild and adapt the cytoskeleton according to the needs of the cell.

A protein called MCAK belongs to a family of motor proteins that can move along microtubules. It generally binds to the ends of the microtubules to shorten them. Previous studies have found that a single MCAK protein binds to another MCAK protein to form a larger molecule known as a dimer. Part of the MCAK protein forms a so-called motor domain, which enables this protein to bind to the microtubules. One end of the protein, known as the C-terminus, controls the activity of this motor domain. However, it is not clear how this works.

Talapatra et al. have now revealed the three-dimensional structure of MCAK's motor domain with the C-terminus using a technique called X-ray crystallography. The experiments show that the C-terminus binds to the motor domain, which promotes the formation of the dimers. A short stretch of amino acids—the building blocks of proteins—in the C-terminus interacts with two motor molecules. This 'motif' is also found in other similar proteins from a variety of animals. However, once MCAK binds to a microtubule, the microtubule triggers the release of the C-terminus from the motor domain. This allows MCAK to bind more strongly to the microtubule.

The experiments also show that the binding of the C-terminus to the motor domain alters the ability of MCAK to associate with microtubules, which encourages the protein to reach the ends of the polymers. Future work is required to see whether other motor proteins work in a similar way.

_Wordeman, 2004_; _Hertzer et al., 2006_; _Zhang et al., 2011_; _Ems-McClung et al., 2013_). The nature and the function of these inter- and intra-molecular interactions within the MCAK dimer are not known.

MCAK is the most potent microtubule depolymerase in the Kinesin-13 family (_Ogawa et al., 2004_). Consequently, its tight regulation is critical for its proper function. Although MCAK depletion causes chromosome segregation defects and lagging chromosomes, MCAK overexpression results in spindle defects and is associated with taxol resistance in cancer cells (_Ganguly et al., 2011a_, _2011b_). The regions that ultimately regulate MCAK targeting and fine-tune the catalytic activity of full-length MCAK lie outside of the motor region. Aurora B phosphorylation at the MCAK N terminus decreases its depolymerase activity (_Andrews et al., 2004_; _Lan et al., 2004_; _Ohi et al., 2004_). CDK1, Plk1, and Aurora A have also been proposed to regulate MCAK activity in vitro (_Zhang et al., 2007_; _Sanhaji et al., 2010_; _Zhang et al., 2011_). Removal of the last 9 amino acids of MCAK from Chinese hamster cells alleviates auto-inhibition of its depolymerase activity by increasing lattice-stimulated ATPase activity, and increases its microtubule binding in vitro (_Moore and Wordeman, 2004_). However, conflicting studies have proposed that the C-terminal tail of MCAK can either inhibit or activate the MCAK depolymerase activity (_Moore and Wordeman, 2004_; _Hertzer et al., 2006_; _Zhang et al., 2011_). Overall, the molecular mechanisms that regulate full-length MCAK activity remain unclear.

Until recently, molecular studies on MCAK have focused on the interaction of the monomeric motor domain with microtubules to dissect the mechanism of MCAK-induced microtubule catastrophe (_Moores et al., 2002_, _2003_; _Ogawa et al., 2004_; _Shipley et al., 2004_; _Mulder et al., 2009_; _Asenjo et al., 2013_; _Zhang et al., 2013_). However, the monomeric motor domain does not function in isolation, as full-length MCAK is a physiological dimer (_Maney et al., 2001_). Recent studies utilizing a FRET probe fused to the neck linker region of MCAK and the C terminus revealed that full-length MCAK switches from a 'closed' to 'open' conformation upon microtubule binding, but the trigger for this conformational change is unknown. MCAK is also thought to adopt a 'closed' conformation at microtubule ends (_Ems-McClung et al., 2013_). The nucleotide state also influences the structure of full-length MCAK and induces a conformational change, as measured by deuterium exchange (_Burns et al., 2014_). These studies suggest that full-length MCAK undergoes large dynamic structural changes during its catalytic cycle and upon binding to microtubules. However, the structure and organization of these flanking regions, and the trigger of the conformational changes remain uncharacterized.

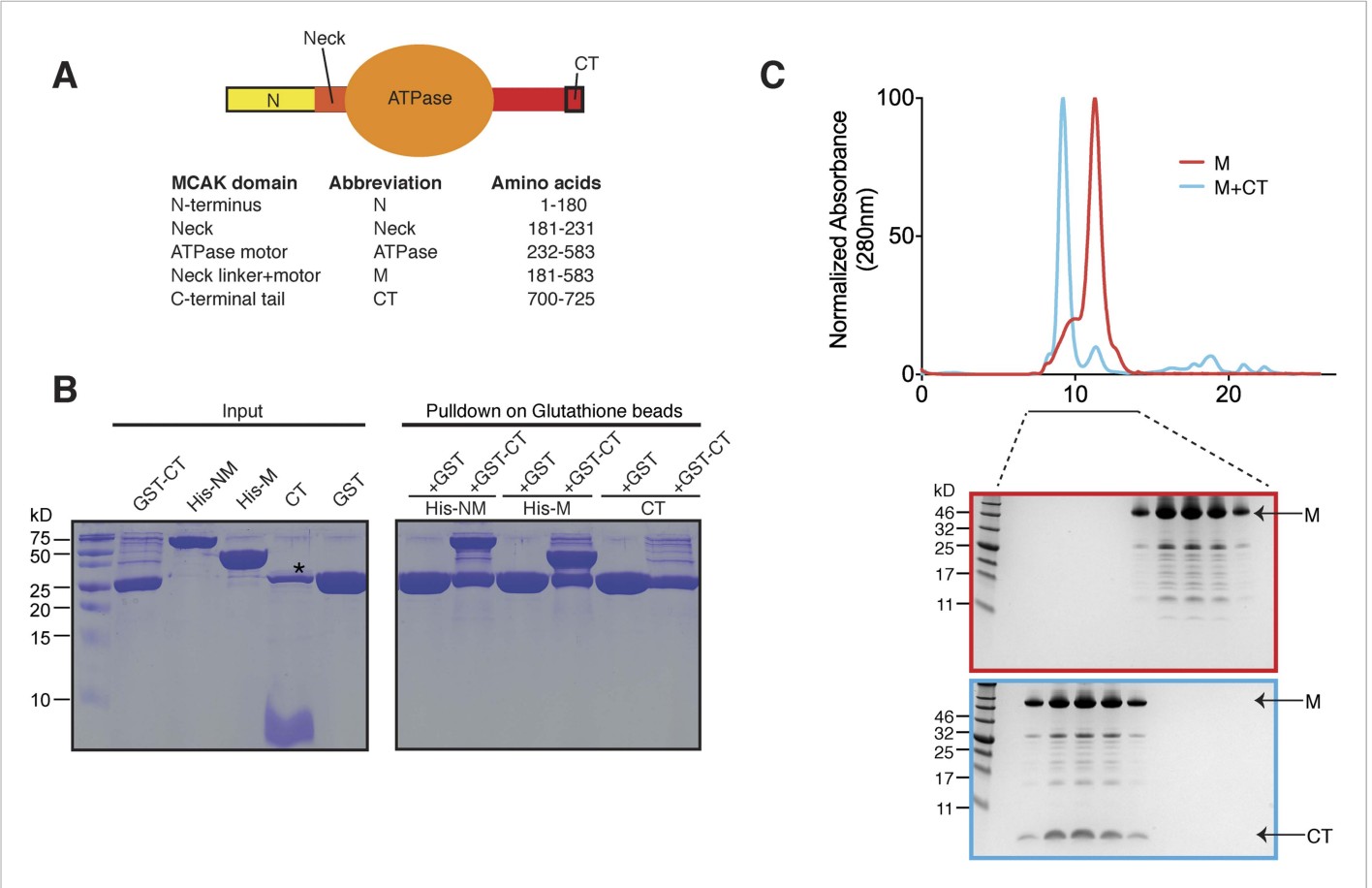

**Figure 1**. The C terminus of MCAK binds to the motor domain. (**A**) Top: schematic diagram showing the different functional domains of full-length MCAK. Bottom: table representing the constructs used and given names. (**B**) Coomassie-stained gel showing a resin-based binding assay for purified His-M, His-NM, and CT domains to either glutathione agarose beads containing GST (as a control) or the GST-CT domain. The star represents residual GST. (**C**) Top, gel filtration elution profile of MCAK motor alone (M, red) and MCAK motor bound to the CT domain (M + CT, cyan). Bottom, coomassie-stained gel showing the size-exclusion chromatography profile of M and M + CT.

Here, we sought to define the molecular basis for the regulation of MCAK by its inter- and intramolecular interactions. Our data establish that the MCAK motor domain binds to a 25 residue peptide from the extreme C terminus, termed the C-terminal tail (CT) domain. The CT domain induces motor dimerization in solution, reminiscent of the self-interaction mechanism of Kinesin-1 with its C terminus (*Kaan et al., 2011*). The crystal structure of the MCAK C-terminal tail bound to the motor domain reveals how the C-terminal domain stabilizes a dimeric MCAK motor configuration. We also show that the MCAK C terminus controls the affinity of full-length MCAK for microtubules and reduces its association with the lattice to ensure maximal recruitment to microtubule ends, where MCAK can act as a depolymerase. When present in solution, the MCAK C-terminus binds to the motor domain. However, upon microtubule binding, the C terminus is displaced from the motor. This step is triggered by the microtubule itself, independently of the E-hook of tubulin and is necessary to allow binding of the motor to microtubules, and stimulate MCAK depolymerase activity. Within the context of the full-length MCAK, this indicates that MCAK undergoes long-range conformational changes driven by its C terminus during its soluble to microtubule-bound transition. Overall, our work presents a new paradigm for kinesin regulation by microtubules rather than their cargos, and provides important insights into the mechanism and regulation of MCAK to control microtubule dynamics and ensure proper genome stability.

## Results

### The MCAK C-terminal region associates with the motor domain

Kinesin-1 is regulated through an auto-inhibitory mechanism whereby one C-terminal tail binds at the interface of the two motors, such that it creates a second point of attachment in addition to the coiled coil region. This limits the head movement of one kinesin with respect to the other (*Stock et al., 1999*; *Hackney and Stock, 2000*; *Kaan et al., 2011*). We sought to test whether the C-terminal tail domain of MCAK, which has been proposed to regulate MCAK activity (*Moore and Wordeman, 2004*), was sufficient to interact with the motor domain. To do this, we expressed the N-terminal and motor region (residues 1–583, termed NM) or the motor region of MCAK along with the neck linker region (residues 181–583, termed M) as His-tagged proteins, and the C-terminal MCAK tail (residues 700–725, termed CT domain) as a GST fusion protein (*Moore and Wordeman, 2004*; *Hertzer et al., 2006*) (*Figure 1A*). Following cleavage and removal of the GST fusion, the CT domain alone was unable to interact with itself through dimerization, based on the absence of binding between the CT domain and GST-CT domain (*Figure 1B*, right lane). We cannot however exclude a very tight interaction between two CT domains. In contrast, the GST-CT domain protein bound to both the NM and M domains of MCAK as a stable complex (*Figure 1B*). In addition, the CT domain bound the motor domain independently of the GST (*Figure 1C*). Together, these experiments reveal that the MCAK CT domain interacts with its catalytic domain in solution.

### The MCAK C terminus binds to two ATPase domains

Above, we demonstrated that the MCAK CT domain interacts with the MCAK motor domain. Full-length MCAK is a dimer in solution (*Maney et al., 2001*). Therefore, we sought to test whether the CT domain interacts with one or two motor domains. To define the stoichiometry of this interaction, we subjected the complex to analytical size-exclusion chromatography. The motor domain alone behaves as a monomer and eluted with an apparent size of ~45 kDa. When the MCAK motor and the CT domain were incubated together and subjected to analytical size-exclusion chromatography, the elution peak shifted to an earlier fraction, suggesting dimerization of the MCAK motor (*Figure 1C*). Using SDS-PAGE analysis, we confirmed that the shift to a larger complex was due to the interaction of the motor domain with the CT domain (*Figure 1C*, bottom). Size-exclusion chromatography coupled with multi-angle light scattering (SEC-MALS) experiments further indicated that, in the absence of the CT domain, over 90% of the motor domain was monomeric (*Figure 2A,B*), with a molecular weight of ~45.3 kDa measured with under 3 kDa accuracy, in agreement with the theoretical molecular weight of ~46.1 kDa (*Figure 2C*). The predicted masses for complexes of one and two CT domains bound to two motor domains are ~94.2 and 97.7 kDa, respectively. The measured molecular weight for the motor-CT domain complex was ~91.5 kDa, suggesting the complex consists of two motors bound to one CT domain (*Figure 2A–C*). Since the CT domain is unlikely to dimerize alone (*Figure 1B*), we conclude that the MCAK CT domain induces the dimerization of the motor domains cooperatively.

We next determined the affinity of the MCAK CT domain for the motor domain using intrinsic fluorescence spectroscopy. We titrated increasing amounts of the CT domain with 1 μM motor domain and measured the corresponding change in the fluorescence intensity of aromatic residues (*Figure 2D*). The change in fluorescence upon peptide binding corresponded to ~1.1 μM affinity of the CT domain for the motor domain, although this measurement does not take into account any existing equilibrium between the motor domains (*Figure 2E*). Overall, this affinity reflects the sum of the dimerization affinity of the motor domains and the affinity of the CT domain for the motor domains, as CT domain binding is cooperative with motor dimerization. Taken together, our data demonstrate that upon binding, the CT domain of MCAK engages with two motor domains.

### The MCAK C terminus binds at the interface between two motor domains

To test how the motor domain of MCAK interacts with the C terminus at the molecular level, we co-crystallized and determined the structure of the motor domain bound to a chemically synthesized peptide corresponding to the CT domain ($_{709}$QLEEQASRQISS$_{720}$) using molecular replacement to

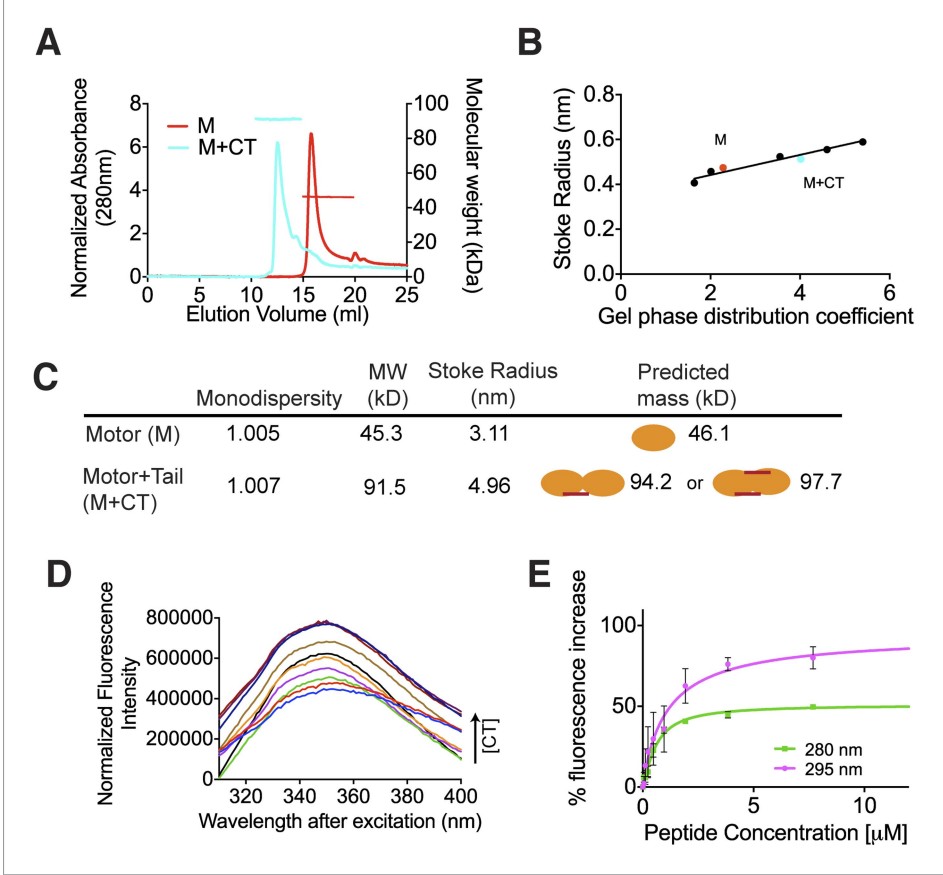

Figure 2. The C terminus of MCAK induces motor domain dimerization. (A) Size-exclusion chromatography elution profiles of motor domain alone (red) and motor domain-CT domain complex (cyan). The horizontal red and cyan lines correspond to SEC-MALS calculated masses for motor domain alone and motor domain-CT domain complex, respectively. (B) Calibration curve for estimation of Stokes radii of motor domain alone (red) and motor domain-CT domain complex (cyan). (C) Table to show the calculated apparent masses and stoke radii of the motor domain-CT domain complex and motor domain alone. The motor domain is drawn in orange and the CT domain in red, to represent the formation of the possible complexes and their predicted size. (D) Steady state intrinsic tryptophan fluorescence emission spectra profile for the titration of the CT domain (CT) ranging from 0 to 15.6 µM, into 1 µM of motor domain after excitation at 295 nm. (E) Effect of the CT domain titration on tryptophan (magenta) and aromatic residue (green) fluorescence quenching of the motor domain. The extent of fluorescence quenching of the motor domain is represented as a percentage of fluorescence change measured for aromatic residues (280 nm) and tryptophan (295 nm) with increasing concentration of wild type CT domain. Relative change in fluorescence after background correction is shown as a function of CT domain concentration. Error bars represent the standard deviation.

a resolution of 3 Å with good stereochemical parameters (*Table 1*, *Figure 3A*). The asymmetric unit contains four molecules of the motor domain assembled into two dimers (chains A and B, C and D) and in a spacegroup distinct from the MCAK motor crystallized alone. The dimerization interface involves packing of the MCAK motor domains along their helix α1 and β3 sheet, close to helix α0 and loop L1, which form the neck linker region (*Figure 3A*, *Figure 3—figure supplement 1A*). From the Fo-Fc electron density map, we could observe interpretable electron density close to the interface between chains A and B and build residues 710 to 716 of the CT domain (*Figure 3B*). We found that a single CT domain binds to both motor domains, close to their neck linker regions. This structural arrangement provides a structural explanation for how the CT domain promotes motors dimerization, reminiscent of the Kinesin-1-tail domain interaction (*Kaan et al., 2011*). The head-to-head motor arrangement is quasi-symmetrical, with the CT domain stabilizing the interface between two motor domains. However, the peptide does not sit on a twofold crystallographic axis and binds

**Table 1.** Data collection, structure determination, and refinement statistics for the X-ray crystal structure of the CT domain of MCAK bound to its motor domain

| Statistics | MCAK motor domain-peptide complex |
|---|---|
| Unit cell dimensions | $a$ = 46.31 Å, $b$ = 245.64 Å, $c$ = 79.40 Å, $\alpha$ = 90.00°, $\beta$ = 95.84°, $\gamma$ = 90.00° |
| Space group | $P2_1$ |
| Molecules per asymmetric unit | 4 |
| Resolution range (Å) | 30.0–3.0 |
| Total reflections | 155983 |
| Unique reflections | 35,146 |
| Completeness (%) | 99.0 (99.2) |
| Multiplicity | 4.4 (4.5) |
| $R_{sym}$ (%) | 9.1 (68.2) |
| $I/\sigma(I)$ | 10.2 (2.0) |
| $R_{work}/R_{free}$ (%) | 26.4/28.6 |
| Wilson $B$ (Å$^2$) | 77.5 |
| Average $B$ (Å$^2$): | |
| Overall | 71.0 |
| Main chain | 72.05 |
| Side chain and solvent | 70.66 |
| Peptide | 56.98 |
| r.m.s.d. bond lengths (Å) | 0.095 |
| r.m.s.d. bond angles (°) | 1.53 |
| Ramachandran plot statistics (%): | |
| Favoured | 87.6 |
| Allowed | 11.7 |
| Outliers | 0.7 |

asymmetrically to the dimer, unlike the Kinesin-1 tail. Our data reveal the molecular basis for the CT domain-induced dimerization of the motor domains, binding along the motor dimer interface to stabilize the complex.

A second potential CT domain-binding site is present in the dimeric motor arrangement for chain A and B. However, it is obstructed by symmetry-related molecules in the asymmetric unit. Chain C and D assemble similarly as dimers, with one potential binding site also obstructed by a symmetry-related molecule. Interestingly, at the second site, the Glu244/C side chain points outwards to the solvent and is incompatible with binding of the CT domain (*Figure 3—figure supplement 1B*). In the CT domain-bound dimer, the side chain of Glu244/A is rearranged and points towards chain B and is stabilized by a salt bridge with Lys286/B. The imidazole ring of His257/B also moves backwards, allowing the CT domain to bind. Thus the motor domains can dimerize, but the CT domain stabilizes this dimeric motor assembly after rearrangement of Glutamate 244 and Histidine 257 (*Figure 3—figure supplement 1B*).

The CT domain contributes the side chains of Glu711 and Glu712, forming charged 'fingers' that dip into the cavity lining the dimeric motor interface to further stabilize the interface (*Figure 3C*). The carboxyl side chain of Glu711 forms a hydrogen bond with His257/B, while the carboxyl group of Glu712 is hydrogen bonded to the amide group of Thr 242/A. Additional hydrogen bonds stabilize the peptide-motor complex through main chain interactions. The backbone amide group of Glu712 is stabilized with the backbone carbonyl group of Ala241/A and the backbone amide of Ser715 hydrogen bonds to the backbone carbonyl group of Cys245/A. Although the hydroxyl group of Ser715 does not interact directly with the motor domain, it does point inwards towards the motor

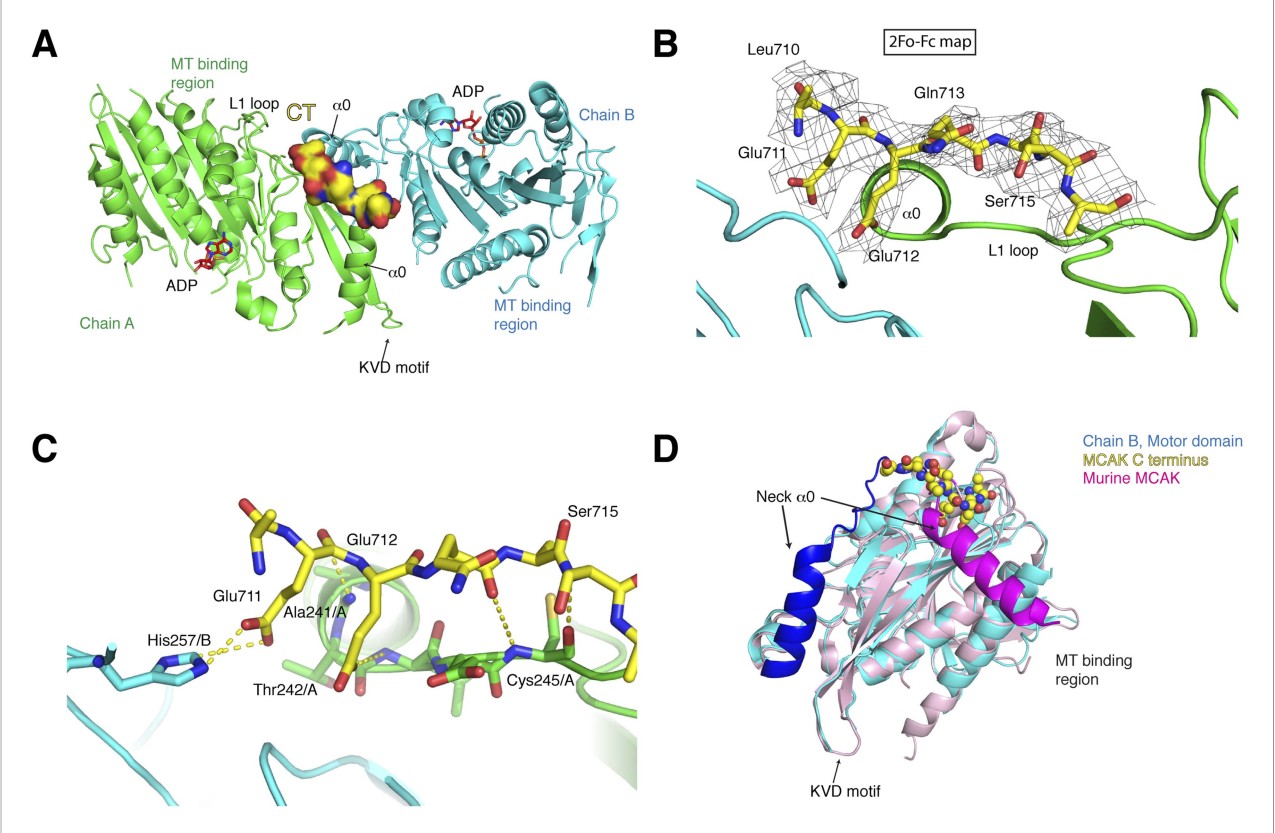

**Figure 3**. Structure of a human motor-CT domain MCAK complex. (**A**) Kinesin motor domain dimers (cyan and green) bound to the CT domain (yellow, spacefill) of MCAK. ADP is in red. (**B**) Motor-CT domain interface showing the electron density map ($2F_{obs} - F_{calc}$), contoured at $\sigma = 1.00$ for the CT domain of MCAK. (**C**) Interactions within the motor-CT domain complex of less than 4 Å are represented by dotted lines. Oxygen and nitrogen atoms are colored red and blue. (**D**) Overlay of the human motor domain and C terminus structure (blue) with the structure of murine MCAK (pink, PDB: 1V8J). The respective neck regions containing the α0 and neck linker are in royal blue and magenta, respectively. The CT domain of MCAK is drawn in yellow as a sphere model with oxygen and nitrogen atoms in blue and red.

The following figure supplement is available for figure 3:

**Figure supplement 1**. Structural analysis of the MCAK motor-CT domain.

domain. The binding of the CT domain occurs far from the P-loop, which forms the ATP binding site, and the switch I and switch II regions (*Figure 3—figure supplement 1C*). In addition, binding of the CT domain does not cause any changes in the ATP binding site or in the overall structure of MCAK (RMSD: 0.863 Å).

Interestingly in our structure, the L1 and the α0 helix part of the neck linker have swung away from the microtubule binding site with respect to the previously published mouse MCAK/Kif2c structure. This suggests that the neck region has conformational flexibility around a hinge region at Arg258, and can adopt at least two states (*Figure 3D*, *Figure 3—figure supplement 1C,D*) (*Ogawa et al., 2004*). Overlaying our structure with the mouse MCAK structure reveals that the conformation of the neck linker region in the mouse MCAK/Kif2c structure does not allow binding of the CT domain, due to steric hindrance (*Figure 3D*, *Figure 3—figure supplement 1D*). The neck region of MCAK has been shown previously to be critical for the depolymerase activity of MCAK (*Maney et al., 2001*; *Ovechkina et al., 2002*). It is therefore possible that disruption of the CT domain-motor interaction allows conformational changes in the neck region that are necessary for catalysis. Taken together, our work reveals that one MCAK CT domain acts directly to stabilize the formation of a dimeric MCAK through an extended interface, where the neck linker lies on the face opposite of the microtubule binding site.

## A conserved motif in the C-terminal region of MCAK is essential for the C terminus-motor interaction

To validate the residues implicated in generating the interface between the MCAK motor domain and CT domain, we generated a series of point mutants to selectively disrupt the binding of the CT domain to the motor domain. Based on our crystal structure and the sequence conservation of the C terminus, we predicted that Glu711 and Glu712 would be critical for the CT domain-motor interaction, whereas Arg716 and Ile718 would not prevent CT domain-motor binding (*Figures 3C, 4A*). As expected from the crystal structure, a $CT_{E711A, E712A}$ domain no longer bound to the motor domain of MCAK, whereas a $CT_{R716A}$ or $CT_{I718A}$ domain bound robustly (*Figure 4C,D*). As revealed in the structure, these two negatively charged glutamic acid residues are critical for the interaction between the CT and motor domains. These two amino acids are conserved from *Drosophila* to human, and are also present in the kinesin-13 family member Kif2a, suggesting that the motor-tail domain interaction is conserved (*Figure 4A,B*).

Interestingly, in addition to these key structural residues, we found that Ser715 in the CT domain is highly conserved across species and is present in the related kinesin, Kif2a, suggesting that this residue could play a role in the tail-motor interaction. Ser715 has been reported to be phosphorylated in vitro by Aurora A and Plk1 (*Zhang et al., 2008, 2011*). In our crystal structure, the hydroxyl group of

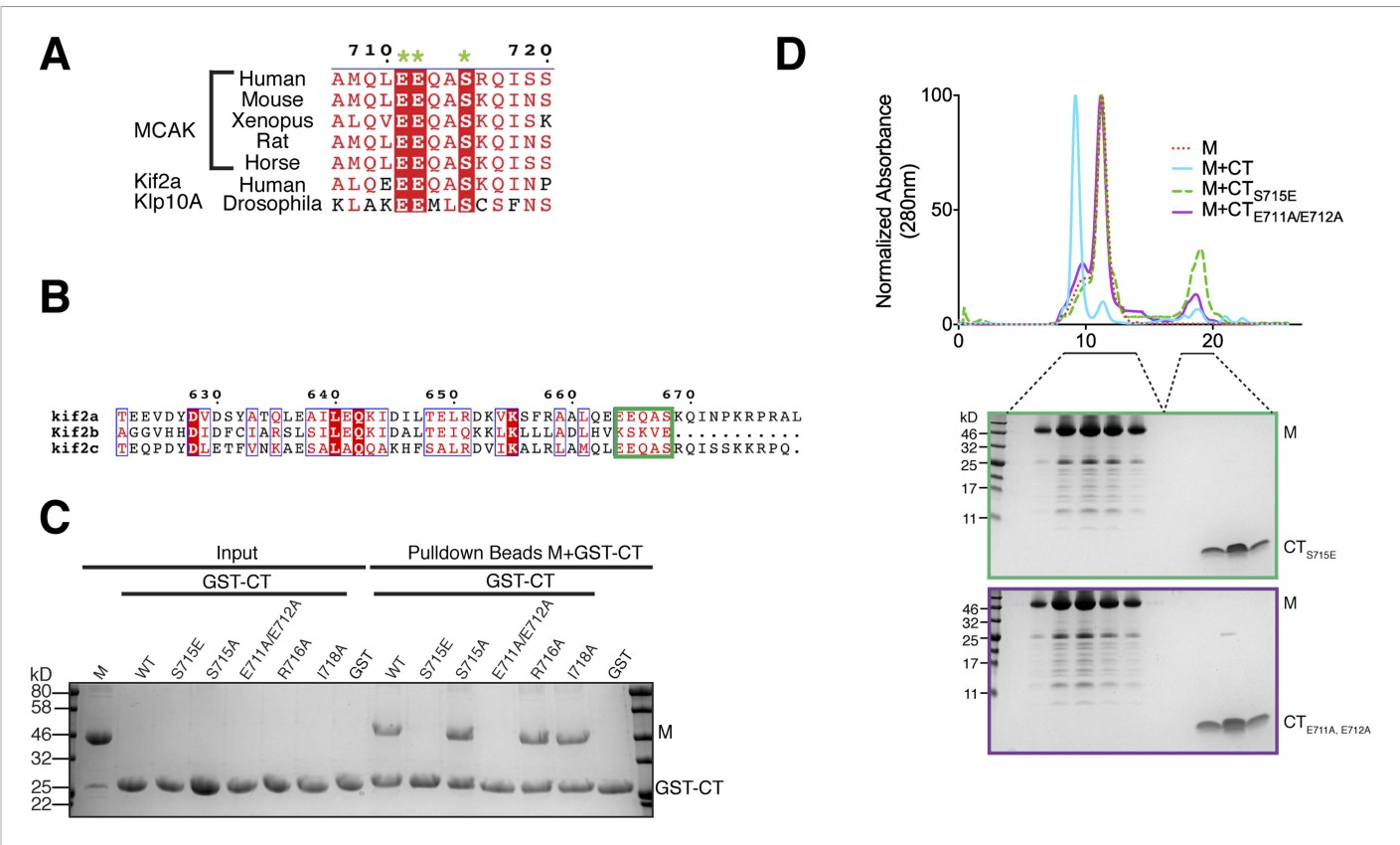

**Figure 4**. Sequence requirement for the formation of a motor-CT tail complex. (**A**) Sequence alignment of the conserved CT domain of MCAK for various species alongside the Drosophila kinesin-13 Klp10A and human Kif2a. The conserved residues are highlighted in red. The three amino acids that are critical for binding to the motor domain are marked with a green star. (**B**) Sequence alignment of the C terminus of human Kif2a, Kif2b, and MCAK/Kif2c. Amino acid numbering is relative to the Kif2a sequence. The MCAK CT domain binding to the motor domain is boxed in green. The sequences were aligned using the program T-coffee (EBI) and formatted with ESPRIPT (*Gouet et al., 1999*). (**C**) Coomassie-stained gel showing a resin-based binding assay using glutathione agarose beads for purified His-M, binding to the GST-CT and GST-CT point mutants. (**D**) Size-exclusion chromatography elution profile of the motor domain alone (red dashes), motor incubated with the CT, $CT_{S715E}$, $CT_{E711A, E712A}$ domains (cyan, green dashes, and purple, respectively). Bottom, coomassie-stained gel showing the size-exclusion chromatography elution of the motor incubated with the $CT_{S715E}$ and $CT_{E711A, E712A}$ domains (green and purple, respectively).

Ser715 is in close proximity to His246 and Glu244. A larger side chain would cause steric hindrance and prevent the CT domain-motor domain association. To test whether the nature of the side chain at position 715 can regulate the interaction between the motor domain of MCAK and its CT domain, we generated GST-CT$_{S715E}$ and GST-CT$_{S715A}$ constructs. Although the GST-CT and GST-CT$_{S715A}$ domains bound to the motor domain, the GST-CT$_{S715E}$ domain did not interact with the motor domain (*Figure 4C*). In addition, the motor domain did not co-migrate with the CT$_{S715E}$ domain by gel filtration (*Figure 4D*). Based on the crystal structure of the CT domain-motor complex, post-translational modification of this residue would destabilize the interaction through electrostatic repulsion and steric hindrance. This demonstrates that the conserved side chains of Glu711, Glu712, and Ser715 are critical for stabilizing the binding of the CT domain to the motor domain. Taken together, our data suggest that the molecular mechanism underlying the interaction between the MCAK C terminus and the motor domain is highly conserved across species.

## Dimerization and depolymerase activity of full-length MCAK are independent of the C terminus

The CT domain induces dimerization of the MCAK motor. To test whether the CT domain was the major dimerization region within MCAK, we generated full-length MCAK$_{S715E}$, in which the CT domain cannot bind to the motor domains. The gel filtration profile of MCAK$_{S715E}$ was similar to MCAK, indicating that MCAK$_{S715E}$ was of similar size to full-length dimeric MCAK in solution (*Figure 5A*). This indicates that there is a second dimerization region within MCAK, independent of the CT domain.

We next asked whether the CT domain affects MCAK depolymerase activity and MCAK function. Removal of the last 9 amino acids at the MCAK C terminus has been reported to increase the lattice-stimulated ATPase activity but not its ATPase activity in solution (*Moore and Wordeman, 2004*). However, conflicting studies have reported that removal of the last 28 amino acids in *Xenopus* MCAK results in a decrease in MCAK depolymerase activity (*Hertzer et al., 2006*). Thus, the role of the CT domain in the context of full-length MCAK remains unclear. The microtubule depolymerase activity of full-length MCAK$_{S715E}$, in which the CT domain can no longer bind to the motor domain, appeared similar to wild type MCAK in microtubule depolymerization assays (*Figure 5B*). However there are limitations to this assay, as we were only able to measure the rate of microtubule depolymerization using cosedimentation assays for a given MCAK concentration rather than examining single MCAK molecules at microtubule ends. It is possible that a change in MCAK microtubule binding affinity will have a counteracting effect on MCAK diffusion rate or the rate of tubulin removal at ends as previously shown (*Cooper et al., 2010*). In this case, the overall depolymerase activity that our assay measures may remain unchanged because the increase in affinity of MCAK for the microtubule lattice may cause a reduction in two-dimensional diffusion and consequently a reduction in microtubule end

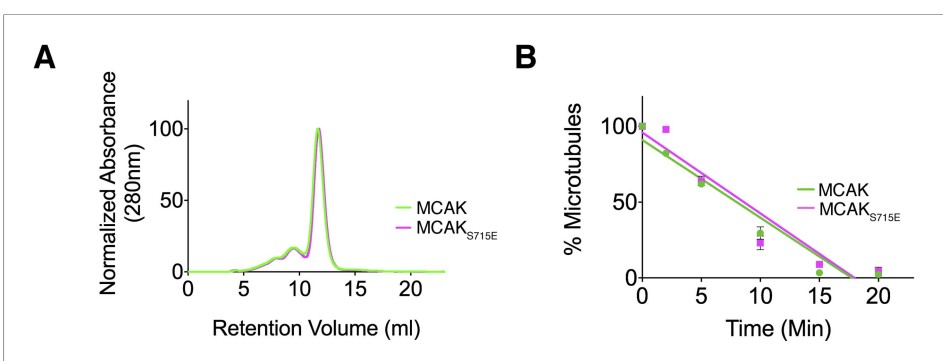

**Figure 5**. Full-length MCAK remains dimeric upon disruption of the motor-CT tail interaction but retains its depolymerase activity. (**A**) Size-exclusion chromatography elution profiles of full-length MCAK (green) and full-length MCAK$_{S715E}$ (magenta). (**B**) Graph plotting the microtubule depolymerase activity of 100 nM MCAK and MCAK$_{S715E}$ by measuring the distribution of 2 μM microtubules in the pellet (P) and free soluble tubulin (S) over time. Error bars represent the standard deviation. Experiments were repeated three times.

targeting. As MCAK and MCAK$_{S715E}$ displayed a similar depolymerase activity in our in vitro depolymerase assay, this raises the possibility that the CT domain acts indirectly as an inhibitor and has an additional distinct cellular function.

## Engineering of a tunable CT domain-motor domain complex

To test the contribution of the CT domain to MCAK activity and function, we designed a system to generate an inducible covalent CT domain-motor complex in vitro based on our structure to control for the displacement of the CT domain from the motor domain. Cys287/A in Loop 1 of the motor domain is in close proximity to the CT domain, with the side chain of Glu712 and the sulfhydryl group of Cys287 pointing towards each other (*Figure 6—figure supplement 1A*). Therefore, we mutated Glu712 to a cysteine to generate a disulfide bridge between the peptide and the motor domain, estimated to be ~3 Å under oxidizing conditions. First, we purified full-length MCAK$_{E712C}$. In presence of reducing agent (DTT), full-length MCAK$_{E712C}$ eluted as one complex, of similar size to MCAK (*Figure 6—figure supplement 2A*). Under oxidizing conditions (without DTT) MCAK$_{E712C}$ ran similarly to MCAK on an SDS-PAGE gel (*Figure 6—figure supplement 2B*). However, we were not able to determine the efficiency of the covalent attachment between Cysteine 712 and Cysteine 287. To test that a covalent linkage had been achieved, we expressed the cleavable GST-CT$_{E712C}$ domain. Under oxidizing conditions (without DTT), the motor and both the GST-CT$_{E712C}$ domain and untagged CT$_{E712C}$ domain formed a covalent complex (*Figure 6—figure supplement 1B,C*). Analytical gel filtration of the GST-CT$_{E712C}$—bound motor complex eluted as a single peak, earlier than the peak for the motor alone (*Figure 6—figure supplement 1D*). However, SDS-PAGE analysis indicated that, within the assembled complex, there was one free motor and one motor covalent bound to the GST-CT$_{E712C}$ domain. This indicates that one CT domain binds to two motors, only one of which is crosslinked (*Figure 6—figure supplement 1B,E,F*). Based on our structural analysis, binding of one CT domain to one of the motors in the dimer would not obstruct the solvent accessibility of the second Cys287. Thus, this experiment suggests that within a CT domain-motor complex, one CT domain binds to two motor domains, consistent with the stoichiometry we determined using SEC-MALS (*Figure 2C*). We also noted that the covalent attachment of the CT domain to the motor domain would also prevent any conformational rearrangement and repositioning of the neck region close to the microtubule-binding interface (*Figure 3D*) and may thus decrease its microtubule depolymerase activity.

## Motor domain binding to the C terminus of MCAK and to microtubules is mutually exclusive

Full-length MCAK has been proposed to undergo large conformational changes upon binding to microtubules, although the underlying mechanism is unclear (*Ems-McClung et al., 2013*; *Burns et al., 2014*). Based on our data, we hypothesized that in solution, the CT domain binds to the motor, but that the CT domain is displaced when the motor binds to microtubules. To test whether MCAK has a reduced ability to bind to microtubules when the CT domain is bound to the motor, we first performed cosedimentation assays with full-length MCAK$_{E712C}$. In the presence of DTT, MCAK$_{E712C}$ bound to microtubules similarly to wild type full-length MCAK. However, under oxidizing conditions (absence of DTT), the affinity of MCAK$_{E712C}$ for microtubules was reduced and a fraction of MCAK$_{E712C}$ did not bind microtubules, even at saturating microtubule concentrations (*Figure 6—figure supplement 2C,D*). This indicates that the binding of the CT domain of MCAK to the motor interferes with MCAK binding to microtubules. To further dissect the effect of the CT domain on the motor domain in the context of microtubules, we performed cosedimentation assays with the CT$_{E712C}$ domain-motor domain complex with increasing concentrations of microtubules. If tubulin within the microtubule is necessary to displace the CT domain and allow binding of the motor to microtubules, we hypothesized that only the non-covalently bound MCAK motor would be able to undergo the conformational change necessary for binding to microtubules, whereas the CT domain-bound MCAK motor (M-CT$_{E712C}$) fraction would be in a locked conformation and would not bind or only bind weakly. Cosedimentation of the motor domain in the presence of the CT$_{E712C}$ domain and DTT was similar to the MCAK motor alone with K$_d$s of 0.44 and 0.64 µM respectively, indicating that the CT$_{E712C}$ domain did not interfere with the motor under reducing conditions (*Figure 6A,B*). Similarly, the addition of DTT did not affect the affinity of MCAK motor in presence of the CT domain (*Figure 6—figure supplement 3A,B*). In contrast, addition of the CT$_{E712C}$ domain to the MCAK motor under oxidizing conditions

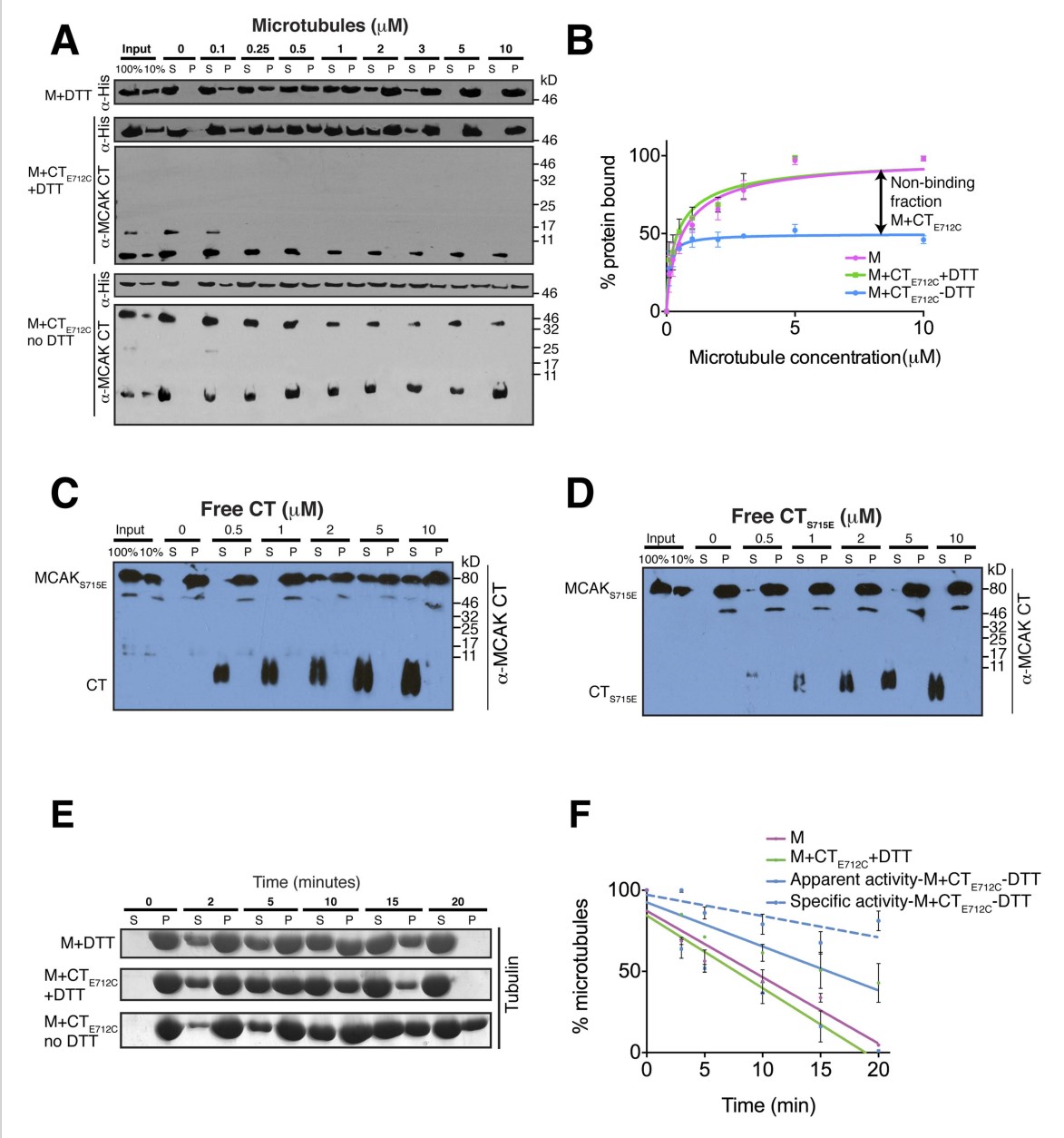

**Figure 6**. The binding of the CT domain to the motor prevents MCAK binding to microtubules and reduces MCAK depolymerase activity. (**A**) Western blot showing the cosedimentation of 50 nM the motor domain of MCAK alone and in the presence of the cleaved $CT_{E712C}$ domain, with and without the addition of DTT to control the formation of the disulphide bridge, at the indicated concentration of microtubules. Detection of the MCAK motor and the CT domain were done using an anti-His and anti-MCAK CT domain antibody, respectively. When the $CT_{E712C}$ domain is covalently bound to the motor domain, we observe free $CT_{E712C}$ domain (~3.5 kD) and motor-bound $CT_{E712C}$ domain (~47 kD) when probing for the CT domain. The $CT_{E712C}$-bound motor remained in the supernatant. (**B**) Graph plotting the microtubule binding activity of the complexes in (**A**) in absence of nucleotide. Data were fitted with a modified Hill equation (*Welburn et al., 2010*). Error bars represent the standard deviation. (**C** and **D**) Western blot showing the cosedimentation of 100 nM full-length $MCAK_{S715E}$ incubated in the presence of 2 μM taxol-stabilized microtubules with increasing concentration of the cleaved free CT (**C**) and $CT_{S715E}$ (**D**) domains. The western blots were probed with the antibody directed against the CT domain. All experiments were repeated three times. (**E**) Coomassie-stained gel showing the microtubule depolymerization activity of 50 nM MCAK motor alone and 50 nM MCAK motor-$CT_{E712C}$ domain in presence and absence of DTT, over time on 2 μM taxol-stabilized microtubules. Free tubulin and microtubule polymers were separated using a cosedimentation assay. (**F**) Graph plotting the quantified microtubule depolymerase activity for conditions in (**E**). The data were fitted with linear regression. The specific depolymerase activity of a covalent MCAK-$CT_{E712C}$ complex was calculated by subtracting the activity of MCAK motor alone, which represents ~50% of the population.

The following figure supplements are available for figure 6:

*Figure 6. continued on next page*

*Figure 6. Continued*

**Figure supplement 1**. Tunable covalent linkage of the CT domain of MCAK to the motor.

**Figure supplement 2**. The affinity of MCAK for microtubules decreases when the CT domain of MCAK is not displaced from the motor.

**Figure supplement 3**. Absence of reducing agent does not affect MCAK motor properties.

reduced the fraction of MCAK bound to microtubules by ∼50%, indicating that half of the CT domain-bound MCAK motor (M-CT$_{E712C}$) sample did not bind to microtubules (*Figure 6A,B*). In these samples, only MCAK motor that was not bound to the CT domain cosedimented with microtubules. Also we did not detect the CT and CT$_{E712C}$ domains in the microtubule-bound, pelleted samples (*Figure 6A*, *Figure 6—figure supplement 3A*). This demonstrates that the binding of MCAK to its C terminal tail region and to microtubules is mutually exclusive.

To further test the effect of the CT domain on MCAK binding to the microtubule lattice, we tested the effect of free CT domain on full-length MCAK$_{S715E}$ in which its own CT domain is unable to bind the motor. We found that the addition of free CT domain decreased the affinity of MCAK$_{S715E}$ for microtubules (*Figure 6C*). In contrast, titration of free CT$_{S715E}$ did not interfere with MCAK$_{S715E}$ binding to microtubules (*Figure 6D*). This indicates that the CT domain specifically competes with microtubules for MCAK binding and effectively reduces the affinity of MCAK for microtubules. Our data suggest that to function as an active depolymerase, MCAK must undergo a large conformational change in which the CT domain of MCAK dissociates from the motor domain and releases the motor domains from each other. In total, our findings demonstrate that the CT domain acts through an allosteric mechanism to prevent MCAK binding microtubules until the CT domain is displaced, thereby enabling the MCAK depolymerase activity.

## The C terminus-motor domain interaction interferes with MCAK depolymerase activity

We next tested the MCAK depolymerase activity when the CT domain is covalently bound to the motor domain. We first tested whether the specific reducing conditions had an effect on MCAK depolymerase activity in the presence of the native CT that could not covalently bind the motor domain (*Figure 6—figure supplement 3C,D*). In both presence and absence of DTT, the MCAK motor could depolymerize microtubules, leading to an increase in free tubulin in the supernatant (S) and a decrease in microtubules in the pellet (P) over time. Next, we incubated the CT$_{E712C}$ domain with the motor in presence and absence of DTT to generate unbound and CT$_{E712C}$-bound MCAK motor. In presence of DTT, the CT$_{E712C}$ domain did not bind the MCAK motor and the depolymerase activity was similar to that of the wild type MCAK motor alone (*Figure 6E,F*). However, in absence of DTT, the CT$_{E712C}$ domain bound-MCAK motor displayed reduced depolymerase activity (*Figure 6E, F*). The activity of this motor-CT$_{E712C}$ domain complex is likely to be lower than that of the observed apparent activity due to the presence of a non-covalently bound MCAK fraction, which functions as a fully active depolymerase (∼45–50%, *Figure 6—figure supplement 1F*). We calculated the specific activity by taking into account the fraction of active MCAK (∼50%, *Figure 6F*–blue dotted line). This shows that the covalent binding of CT$_{E712C}$ to MCAK motor strongly inhibits the depolymerase activity of MCAK. Overall, this demonstrates that the displacement of the CT domain is necessary for the full microtubule depolymerization activity of MCAK.

## The tubulin lattice triggers the release of the C terminus from the motor domain

Above, we found that the displacement of the CT domain is required for MCAK motor association with the microtubule (*Figure 6*). However, the molecular mechanism that triggers the displacement of the CT domain from the motor was unclear. To test whether the negatively charged E-hook of tubulin or the lattice itself triggers the removal of the CT domain from the motor, we performed cosedimentation assays of the motor bound to the CT domain with microtubules in absence of the tubulin tails. To test this, we treated microtubules for 10 and 120 min with subtilisin to remove the

C-terminal tails of β and α/β-tubulin, respectively (*Figure 7—figure supplement 1A*). Cosedimentation of the motor-CT domain complex in presence of subtilisin-treated microtubules revealed that the CT domain was displaced from the motor and remained in the supernatant, while the motor domain bound with a high affinity to the tubulin lattice (Kd = 0.2 μM) (*Figure 7—figure supplement 1B,C*). Removal of the α-tubulin tail did not further modify the affinity of the motor domain for microtubules that also lacked the β-tubulin tail. Taken together, the microtubule lattice itself rather than the acidic tails of tubulin trigger the release of the CT domain from the motor.

### The C terminus of MCAK and the E-hook of tubulin both reduce the apparent affinity of MCAK for microtubules

Removal of the entire C-terminal domain of MCAK has been shown to increase the affinity of MCAK for microtubules and prevent plus end targeting, although the mechanism is not defined (*Moore and Wordeman, 2004*; *Moore et al., 2005*). In addition, we found that the CT domain reduces the ability of full-length MCAK to bind to microtubules (*Figure 6C*). Therefore, we predicted that full-length MCAK$_{S715E}$, in which the CT domain is unable to interact with the motor domains would have a higher affinity for microtubules than wild type full-length MCAK. To test this, we measured the affinity of full-length wild type MCAK and MCAK$_{S715E}$ for microtubules using a cosedimentation assay. We found that MCAK$_{S715E}$ showed a 10-fold increase in the apparent affinity for microtubules compared to wild type MCAK (~0.2 μM and 1.5 μM, respectively; *Figure 7A*). MCAK has been reported previously to

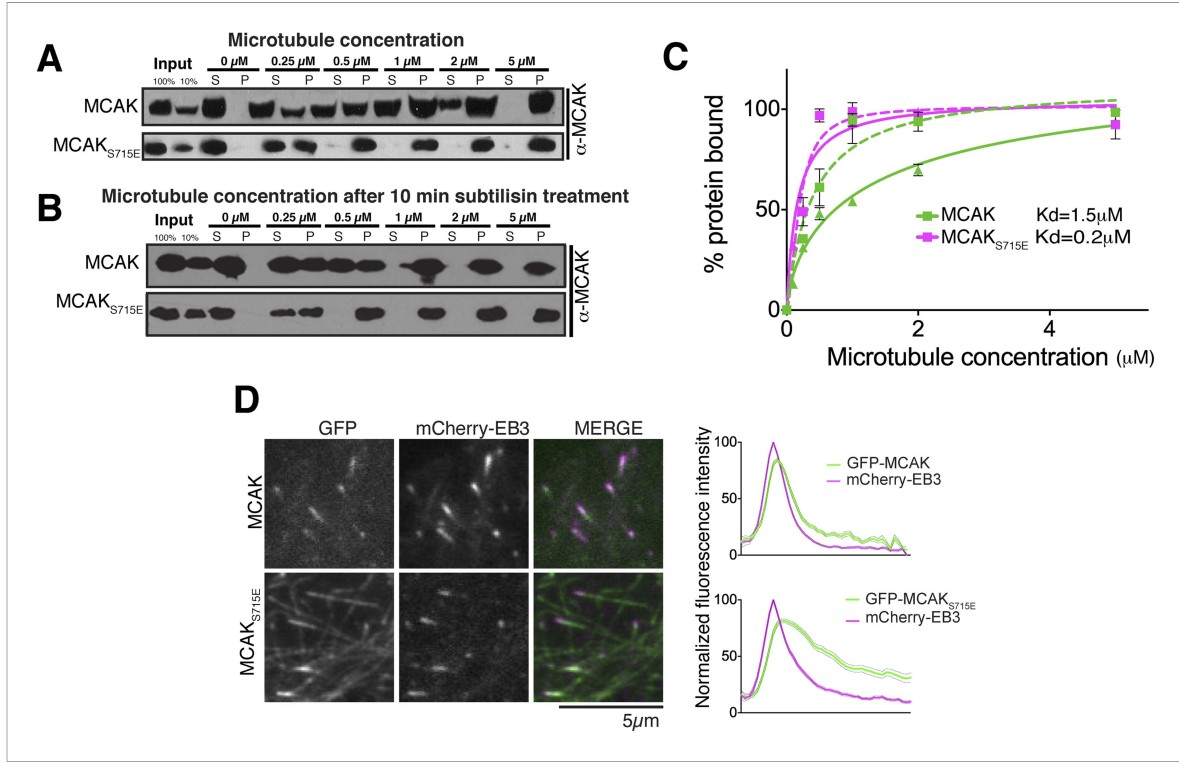

**Figure 7**. The CT domain reduces the affinity of MCAK to microtubules. (**A** and **B**) Western blot showing the cosedimentation of 50 nM MCAK with microtubules at the indicated concentrations. In panel **B**, the microtubules have been treated with subtilisin for 10 min prior to the cosedimentation assay. (**C**) Graph plotting the average microtubule binding activity of MCAK and MCAK$_{S715E}$ in absence of nucleotide. The dashed and full curves correspond to subtilisin-treated and untreated microtubules, respectively. The data were fitted using a modified Hill equation. Error bars represent the standard deviation. (**D**) Representative images of HeLa cells transiently transfected with mCherry-EB3 and GFP-MCAK or GFP-MCAK$_{S715E}$, alongside the respective average normalized fluorescence intensity linescan profiles at microtubule plus tips. Grey shading of the linescans represents the standard error.

The following figure supplement is available for figure 7:

**Figure supplement 1**. The displacement of the CT domain from the motor is triggered by the microtubule lattice but is independent of the E-hook of tubulin.

bind to microtubules lacking the acidic tails (*Niederstrasser et al., 2002*; *Helenius et al., 2006*). However, these studies indicated that the ability of MCAK to diffuse on the lattice was reduced in this case. To test the effect of the acidic C-terminal tails of tubulin on MCAK binding and on the function of the CT domain, we tested the affinity of MCAK and MCAK$_{S715E}$ for subtilisin-treated microtubules (*Figure 7B*). Removal of β-tubulin C termini increased the affinity of full-length MCAK, whereas the affinity of full-length MCAK$_{S715E}$ for microtubules remained comparably high (*Figure 7C*). This indicates that both the CT domain of MCAK and the C termini of tubulin cooperate to reduce the affinity of MCAK for microtubules and ensure that MCAK does not become trapped on the lattice, away from its microtubule ends substrate.

MCAK utilizes weak tethering to diffuse on the negatively charged C-terminal tails of the microtubule lattice (*Helenius et al., 2006*). The neck region was originally proposed to promote MCAK diffusion, similarly to Kif1a (*Thorn et al., 2000*; *Wang and Sheetz, 2000*; *Ovechkina et al., 2002*; *Helenius et al., 2006*). The idea that the neck was the electrostatic tether supporting E-hook mediated diffusion was subsequently disproven (*Cooper et al., 2010*). Thus to date the regions of MCAK responsible for diffusion remain unclear. Based on the CT domain controlling the affinity of MCAK for microtubules redundantly with the C-terminal tail of tubulin, we hypothesized that this electrostatically charged CT region may play also a role in MCAK diffusion on the lattice and targeting to microtubule ends (*Figure 8*). To test whether the CT domain of MCAK controls the targeting of MCAK by decreasing MCAK affinity for the microtubule lattice, we examined the localization of full-length MCAK and MCAK$_{S715E}$ in HeLa cells. GFP-MCAK$_{S715E}$ targeted weakly to microtubule plus ends but also accumulated on the microtubule lattice (*Figure 7D*), confirming our in vitro observation (*Figure 7A,C*). In contrast, GFP-MCAK was robustly targeted to microtubule ends and co-localized with mCherry-EB3. Future work will address whether the CT domain is the main region providing direct lattice diffusion properties to MCAK through electrostatic interactions. In total, these data suggest that the CT domain reduces the affinity of MCAK for microtubules and may be the electrostatic tether that allows MCAK specific targeting to microtubule ends.

## Discussion

MCAK is a powerful microtubule depolymerase, whose activity must be tightly regulated through phosphorylation and self-interaction. Our results reveal a regulatory paradigm for the Kinesin-13 microtubule depolymerases, which are functionally and structurally distinct from processive kinesins. Previously, the molecular organization of full-length dimeric kinesin depolymerases and the inhibitory mechanisms for kinesin depolymerases were unclear. Here, we show that in solution, the C terminus of MCAK interacts with the two motor domains through long-range interactions. Binding of the CT domain and microtubules to the motor is mutually exclusive. While the acidic tails of tubulin control

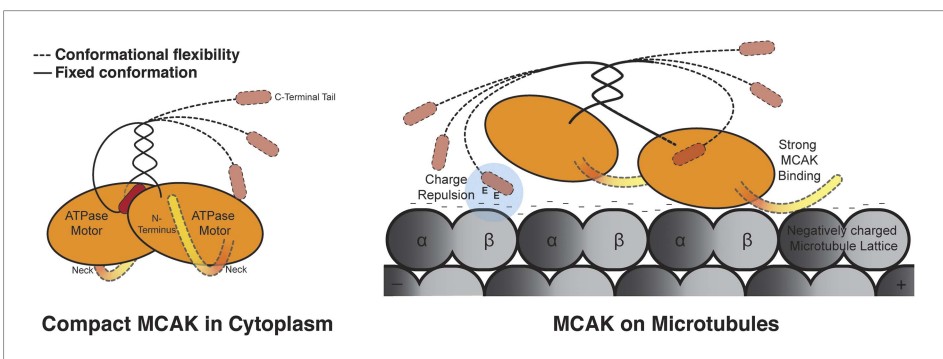

**Figure 8**. Model for MCAK conformation in solution and when bound to microtubules. MCAK has a compact structure in solution, with one C terminus binding at the interface between two motor domains. MCAK can bind to microtubules through the microtubule-binding region, which allosterically triggers release of the C-terminus of MCAK. The motor domains can then efficiently bind to and depolymerize the microtubule end, through possible repositioning of the neck linker region. Both the C terminus of MCAK and the negatively charged E-hook of tubulin, reduce the binding of MCAK to microtubules, enabling MCAK to diffuse efficiently to microtubule ends.

the affinity of MCAK for microtubules, they are not necessary for the displacement of the MCAK CT domain from the motor. The tubulin subunit itself triggers the removal of the CT domain from the motor, most likely through a conformational change within the microtubule-binding region in the motor domain. Disruption of this interaction causes MCAK to bind more strongly to microtubules, which leads to the accumulation of MCAK along the microtubule lattice and is disadvantageous for a microtubule-depolymerizing enzyme that acts at microtubule ends. Removal of the tubulin C termini also increases the affinity of MCAK for the microtubule lattice. Therefore, we propose that both the CT domain of MCAK and the C termini tubulin have important functions to reduce the affinity of MCAK for the microtubule lattice and facilitate MCAK diffusion, in part through electrostatic repulsion, in agreement with previous observations on a tailless MCAK (*Moore and Wordeman, 2004*; *Helenius et al., 2006*). The CT domain of MCAK has a predominant effect on controlling MCAK affinity through intramolecular interactions, possibly through weak intramolecular interactions with the motor not engaged with the lattice or by interfering with the E-hook of tubulin. Taken together, both the CT domain of MCAK dimers and the acidic tails of tubulin effectively contribute to efficient microtubule lattice engagement, plus tip targeting, and activation of the depolymerase (*Helenius et al., 2006*). This model could explain why MCAK activity is stimulated by the microtubule lattice and requires both its CT domain and the C terminus of tubulin for optimal activity (*Niederstrasser et al., 2002*; *Helenius et al., 2006*; *Cooper et al., 2010*).

The MCAK C-terminal binding motif 'EEXXS' is conserved across species from *Drosophila* to Human and is present in the kinesin-13 family member Kif2a, suggesting that this regulatory targeting mechanism is highly conserved (*Cameron et al., 2006*). Interestingly, the C terminus of the Kinesin-13 member Kif2b diverges dramatically from Kif2a and MCAK (*Figure 4B*). Kif2b binds to Cep170 through its C terminus to enhance its targeting to the spindle. The Kif2b C-terminal tail regulates kinesin activity through an alternate mechanism based on an association with binding partners (*Welburn and Cheeseman, 2012*).

Our work reveals that MCAK undergoes long-range conformational changes during its transition from soluble to microtubule-bound state. The extreme C terminus of MCAK binds to the motor domain in solution and this interaction is abrogated upon MCAK binding to microtubules. This implies that a major microtubule-induced conformational change in MCAK occurs by disrupting the regulated interaction of the motor with the CT domain, which is triggered by the microtubule lattice itself. This event may also allow and require rearrangement of the neck region, which can swing into two distinct conformations on opposite faces of the MCAK motor domain. Recent work reported that MCAK undergoes long-range conformational changes upon binding to microtubules based on FRET (*Ems-McClung et al., 2013*), although the nature of the change was unknown. Aurora B phosphorylation of the neck region has been proposed to control the long-range interactions with a C-terminal non-motor region of MCAK, however the molecular basis for this regulatory mechanism was lacking. Recent low resolution studies using deuterium-exchange and mass spectrometry also indicated that the C terminus of MCAK within the context of the full-length MCAK is more stable in solution than in the presence of microtubules (*Ems-McClung et al., 2013*; *Burns et al., 2014*). Thus, our studies reveal the molecular basis for this microtubule-induced change in conformation.

An increasing number of kinesins also appear to be regulated by self-interactions (reviewed in *Welburn, 2013*). Kinesin-1, Kif17, and CENP-E can each undergo self-inhibition in solution to limit squandering of ATP (*Coy et al., 1999*; *Friedman and Vale, 1999*; *Hackney and Stock, 2000*; *Espeut et al., 2008*). We currently only have molecular insights into the inhibitory mechanism for Kinesin-1, where one C-terminus binds at the interface between two motor domains to inhibit the molecular motor (*Hackney et al., 2009*; *Kaan et al., 2011*). Here, we demonstrate that certain features of the molecular inhibitory mechanism for processive kinesins can be extended to depolymerizing kinesins despite their different structural arrangement but that the function of this self-interaction is distinct. In both cases, the C terminus acts allosterically and stabilizes a motor domain dimer through a second dimerization site, distinct from the major dimerization domain (*Kaan et al., 2011*). However, in the structure of the Kinesin-1 tail complex, the tail binds on a twofold symmetry axis utilizing two ionic interactions. The tail binds symmetrically and in both directions on the motor around a twofold symmetry axis. We found that the MCAK CT domain binds asymmetrically with multiple interactions with the motor domain and adopts only one potential orientation (*Figure 3*). Once this occurs, the interaction of the tail with the MCAK motor displays a reduced affinity for microtubules, similarly to Kinesin-1. However, while Kinesin-1 auto-inhibition reduces ATPase activity, the C terminus of MCAK

does not interfere with ATP hydrolysis in solution (*Hackney and Stock, 2000*; *Moore and Wordeman, 2004*). In addition, unlike motile kinesins, the alleviation of MCAK auto-inhibition is not stimulated by cargo proteins, but rather by the microtubule lattice itself, although the removal of the tail is in both cases electrostatically-driven (*Stock et al., 1999*). In total, our work reveals that regulation of self-interactions in the kinesin superfamily emerges as a conserved feature, but that the nature of their regulation is distinct between processive and depolymerizing kinesins. Future structural work on MCAK will help us understand how this potent non-canonical kinesin functions in vivo.

## Materials and methods

### Protein expression and purification

His-MCAK (1–725), His-MCAK$_{S715E}$, and His-MCAK$_{E712C}$ were cloned in the pFL vector and subsequently used for Sf9 cell expression using the BEVS baculovirus expression system and protocol (*Fitzgerald et al., 2006*). Full-length MCAK proteins were purified as described earlier (*Moore and Wordeman, 2004*). His-MCAK (183–583, M) and His-MCAK (1–583, NM) were subcloned in pET3aTr vector. For the CT domain of MCAK (700–725), two long primers with BamHI and XhoI restriction sites; forward: 5′ – CCCGGATCCATCAAGGCCT

TGCGCCTGGCCATGCAGCTGGAAGAGCAGGCTAGCAGACAAATAAGCAGCAAGAAACGGCCCCA GTGACTCGAGCCC – 3′ and reverse: 5′ – GGGCTCGAGTCACTGGGGCCGTTTCTTGCTGCTTATTTGTC TGCTAGCCTGCTCTTCCAGCTGCATGGCCAGGCGCAAGGCCTTGATGGATCCGGG – 3′ were first annealed together as double stranded DNA. The insert was then ligated into pGEX 6p1 (GE Healthcare Life Sciences, UK). For the MCAK CT$_{EEEEE}$, a gene encoding the C terminus of MCAK was synthesised by LifeTechnologies and was subcloned into the MCAK vector, described in *Welburn and Cheeseman (2012)*. Amino acids 716 and 718–721 were mutated to glutamates.

Protein expression was induced by addition of 0.5 mM IPTG to BL21(DE3) Codon plus cells transformed with respective constructs at OD$_{600}$ of 0.7–0.8 for 16 hr at 18°C. Cells were lysed by sonication in lysis buffer (50 mM Hepes, pH 7.4, 200 mM NaCl, 1 mM MgCl$_2$, 1 mM PMSF, 1 mg/ml DNaseI, 2 mg/ml lysozyme, 10 mM Imidazole) and clarified at 20,000 rpm for 1 hr at 4°C. His-tagged and GST-tagged proteins were subsequently purified using Ni-NTA–agarose beads and glutathione-sepharose beads, respectively (GE Healthcare Life Sciences, UK)) according to the manufacturer's guidelines. MCAK constructs containing the motor domain were eluted with elution buffer (50 mM Hepes, pH 7.4, 200 mM NaCl, 1 mM MgCl$_2$, 1 mM ATP, and 300 mM imidazole). Cleavage of the GST tag was performed using the GST-3C protease overnight at 4°C. Proteins were further purified using gel filtration chromatography pre-equilibrated in gel filtration buffer (For full-length MCAK: 100 mM HEPES, pH 7.3, 200 mM NaCl, 200 mM KCl, 1 mM DTT, 1 mM MgCl$_2$, 1 mM Na-EGTA, 1 mM ATP; for the motor domain constructs: 50 mM HEPES, pH 7.2, 150 mM NaCl, 1 mM DTT, 1 mM MgCl$_2$, 1 mM Na-EGTA, 1 mM ATP; for CT domain constructs: 50 mM HEPES, pH 7.2, 150 mM NaCl, 1 mM DTT, 1 mM MgCl$_2$, 1 mM Na-EGTA). Analytical gel filtration chromatography was performed using either a Superdex 75 or a Superose 6 10/300 GL column (GE Healthcare, UK). To purify the CT domain alone, we cleaved GST and CT after gel filtration and a concentration step, and performed a glutathione affinity-purification third step to remove the GST and collect the CT domain. The CT domain was then further purified by separating it from remaining GST using a concentrator with a 3 kD-cutoff. Protein concentrations were determined with a combination of Bradford protein assays and densitometry of Coomassie-stained gels relative to a BSA standard. To visualize both the motor and CT domains on protein gels, 16% Tricine gels were used, according to the manufacturer's instructions (Invitrogen, Life Technologies, Paisley, UK).

### Binding studies using intrinsic aromatic amino acid fluorescence

MCAK motor domain was pre-treated with spectroscopy buffer (100 mM HEPES, pH 7.4, 150 mM NaCl) supplemented with 5 mM EDTA to remove any bound ADP. The protein was then desalted into spectroscopy buffer using a Disposable PD-10 Desalting Columns (GE Healthcare Life Sciences, UK). The experiment was performed with a modified protocol as previously described (*Chadborn et al., 1999*). A Cary 2200 spectrophotometer was used to measure absorption spectra; fluorescence was measured using an ISS K2 spectrofluorometer at 25°C. The intrinsic fluorescence of tryptophan and aromatic amino acids after excitation at 295 nm and 280 nm, respectively was recorded through an Ealing 340 nm centre-wavelength filter. The emission spectra were measured from 300 to 400 nm. The

motor domain was diluted to 1 µM in spectroscopy buffer. First, the emission spectrum for the motor domain alone was recorded. Then the following concentrations of CT domain peptide, cleaved from GST and further purified, were titrated: 0, 60, 120, 240, 480, 960, 1920, 3840, 7680, and 15,660 nM. The starting volume was 3 ml before peptide addition and was never increased more than 1% to negate any effect on fluorescence measurements. Because of the presence of GST (5% of the total peptide) all measurement was corrected with measurement of buffer containing the same concentration of peptide. The change in fluorescence was calculated after they were normalized against each concentration of the CT domain alone in the spectroscopy buffer, to correct for non-specific fluorescence.

## Microtubule cosedimentation assays

Full-length MCAK complex was diluted in S buffer (50 mM NaCl, 20 mM Hepes pH 7.0) to 50 nM, in absence of nucleotide to prevent MCAK-dependent microtubule depolymerisation. To assemble an MCAK motor-CT domain complex, 50 nM of the motor domain and 100 nM of the CT domain were used. Microtubule binding assays were performed as described (*Cheeseman et al., 2006*) using equal volumes of taxol-stabilized microtubules in BRB80 and MCAK in S buffer. MCAK was quantified using anti-MCAK antibody against the C terminus of MCAK ($_{709}$QLEEQASRQISS$_{720}$), generated by GL Biochem (Shangai) Ltd (China) or anti-His antibody to probe for the motor domain alone (GE Healthcare Lifesciences, UK). The data from at least three independent experiments were fitted to a modified Hill equation to determine the apparent $K_d$.

## Microtubule depolymerisation assay

Microtubule depolymerization assays were performed essentially as described previously (*Hertzer et al., 2006*). MCAK was diluted to 100 nM in S-buffer containing 1 mM DTT and 2 mM Mg-ATP. For depolymerization assays of the motor domain in presence of the CT domain after GST cleavage and removal, 50 nM of motor domain and 100 nM of the CT domain were used. The microtubule depolymerization assay was initiated by the addition of 2 µM taxol-stabilized microtubules to a reaction buffer containing MCAK. Reactions were incubated at room temperature with increasing times and followed by centrifugation to separate microtubules from free tubulin. The data are represented as mean ± SD from three independent experiments.

## Covalent attachment of the CT domain to the motor domain

The GST-CT$_{E712C}$ and CT$_{E712C}$ domains were cross-linked to the motor domain by incubating them with the motor domain in a buffer containing 100 mM HEPES, pH 7.4, 150 mM NaCl but lacking DTT for 1 hr at 4°C. As a negative control the motor domain and the CT$_{E712C}$ domain were incubated in the identical buffer, supplemented with 5 mM DTT.

## Subtilisin treatment of microtubules

Tubulin (5 mg/ml) was first polymerized into MTs in the presence of 1 mM GTP and gradual addition of 0.05 µM, 0.5 µM and 2 µM taxol for 1 hr at 37°C. The polymerized MTs were then treated with 100 µg/ml subtilisin and incubated at 37°C for 10 min to cleave β-tubulin tails and 120 min to cleave both α- and β-tubulin tails. Each reaction was then terminated with the addition of 3 mM PMSF. DM1A (Abcam, UK) and c-terminal β-tubulin (Sigma, UK) were used to detect the α- and β-tails, respectively by western blotting. The subtilisin treated microtubules were then pelleted at 28°C in a TLA100 rotor at 80,000 rpm for 10 min and the microtubule pellets were resuspended in warm BRB80 buffer to obtain subtilisin treated microtubules. The cosedimentation assays were then performed as described before.

## Size-exclusion chromatography coupled to multi-angle light scattering

Size-exclusion chromatography with on-line multi-angle light scattering (SEC-MALS) was performed using a GE Superdex 200 10/300 GL column on an ÄKTA FPLC system. MALS measurements were performed using a MiniDAWN in-line detector (Wyatt Technology, Santa Barbara, CA, USA). MCAK motor domain and C terminus were at 2 mg/ml in 100 mM HEPES, pH 7.2, 150 mM NaCl. Protein concentration was monitored using a UV monitor at 280 nm and a refractive index detector was set at 690 nm (Optilab DSP, Wyatt Technology, Santa Barbara, CA, USA). Data were analyzed using Astra software (Wyatt Technology, Santa Barbara, CA, USA) using the refractive index detector and a refractive index increment (dn/dc) value of 0.185 ml/g. Gel phase distribution coefficients ($K_{av}$) were determined from the

equation $K_{av} = (V_e - V_o)/(V_t - V_o)$, where $V_e$, $V_o$, and $V_t$ represent the elution volume of the protein of interest, the column void volume and the total bed volume of the column, respectively.

## Crystallization of MCAK motor domain and tail complex

1 mM MCAK motor domain (PDB:2HEH, Addgene, Cambridge MA, USA) was incubated with the CT peptide $_{709}$QLEEQASRQISS$_{720}$ (China peptides Co, Ltd, China) in a ratio of 1:2 for 1 hr at 4°C before setting up crystallization trials. Elongated rectangular crystals appeared by vapor diffusion after two days in sitting drops using 24% wt/vol PEG 1500 and 20% vol/vol Glycerol as a precipitant. Crystals were grown in MRC 2 Well Crystallization Plate (Hampton Research, Aliso Viejo, CA, USA) at 19°C. Crystals were cryoprotected in a solution containing 28% wt/vol PEG 1500 and 30% vol/vol Glycerol and flash-frozen in dry liquid nitrogen.

## Structure determination, refinement, and model quality

Diffraction data were recorded at Diamond Light Source on beamline ID24 at 100 K. Data were processed using XDS package (*Kabsch, 2010*) and SCALA operated through the CCP4 suite GUI (*Collaborative Computational Project, Number 4, 1994*). The structure of the MCAK motor-tail complex was solved by molecular replacement using the program MOLREP. The MCAK motor domain structure (PDB code: 2HEH) was used as a search model. Structure refinement was performed using Refmac5 and Phenix (*Adams et al., 2010*). Model quality statistics are summarized in *Table 1*. Figures were prepared using PyMOL (*Delano, 2002*).

## Accession number

The final model and the structure factor amplitudes have been submitted to the Protein Data Bank under the accession code 4UBF.

## Cell culture and fluorescence microscopy imaging

Transfection of GFP-MCAK and mCherry-EB3 constructs in HeLa cells was performed using Effectene (Quiagen, Dusseldorf, Germany) according to manufacturer's instructions. Images were acquired on a Nikon TIRF inverted microscope system with a perfect focus, with a 100× TIRF Apo 1.49 objective (Nikon, UK) using an Andor Zyla technology Scmos camera. Imaging was carried out at 37°C. Images were analyzed using ImagePro software and OMERO. Linescan averages were calculated from over 100 comets.

## Acknowledgements

We thank Adele Marston, Iain Cheeseman, Steve Royle and the Welburn lab and the Cell cycle floor meeting participants for discussions and critical reading of the manuscript and Atlanta Cook for advice. We thank Andrew Carter for help with purifying tubulin. We thank Dr Martin Wear and Dr Liz Blackburn at the EPPF for their input in the analysis of SEC-MALS and intrinsic fluorescence quenching experiments. We thank Diamond Light Source for access to beamline I24 (MX9487) that contributed to the results presented here. JW is supported by a CRUK Career Development Fellowship (C40377/A12840). BH is supported by a Wellcome Trust Studentship (099843).

## Additional information

### Funding

| Funder | Grant reference | Author |
| --- | --- | --- |
| Cancer Research UK | C40377/A12840 | Julie PI Welburn |
| Wellcome Trust | 099843 | Bethany Harker |

The funders had no role in study design, data collection and interpretation, or the decision to submit the work for publication.

### Author contributions

SKT, Conception and design, Acquisition of data, Analysis and interpretation of data; BH, Acquisition of data, Analysis and interpretation of data; JPIW, Conception and design, Acquisition of data, Analysis and interpretation of data, Drafting or revising the article

## Additional files

### Major datasets

The following dataset was generated:

| Author(s) | Year | Dataset title | Dataset ID and/or URL | Database, license, and accessibility information |
|---|---|---|---|---|
| Welburn JPI, Talapatra SK | 2015 | HsMCAK motor domain complex | http://www.pdb.org/pdb/search/structidSearch.do?structureId=4UBF | Publicly available at RCSB Protein Data Bank (4UBF). |

The following previously published dataset was used:

| Author(s) | Year | Dataset title | Dataset ID and/or URL | Database, license, and accessibility information |
|---|---|---|---|---|
| Wang J, Shen Y, Tempel W, Landry R, Arrowsmith CH, Edwards AM, Sundstrom M, Weigelt J, Bochkarev A, Park H | NA | Crystal Structure of the KIF2C motor domain (CASP Target) | http://www.pdb.org/pdb/explore/explore.do?structureId=2HEH | Publicly available at RCSB Protein Data Bank (2HEH). |

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
