## [Decision Letter]

Thank you for sending your work entitled “The C-terminal region of MCAK controls
its structure and activity through a conformational switch” for consideration at
*eLife*. Your article has been evaluated by John Kuriyan (Senior
editor) and two reviewers. The editor has concluded that the paper may become suitable
for publication in *eLife* if the issues raised by the referees can be
addressed suitably. The reviewers, who also reviewed your original submission, have
discussed their concerns with each other and the editor, and the review report merges
all of the comments. The review report is appended below.

Review:

The authors present the crystal structure of the motor domain of the microtubule
depolymerase MCAK in complex with a C-terminal peptide. The structure reveals that this
peptide promotes dimerization of the motor-domain and regulates the conformation of
MCAK's neck linker region. The structural data are well confirmed by biophysical
methods and mutant studies.

The second part of the paper uses this structure to work out the role of the C-terminal
peptide. The authors use covalent attachment of the C-terminal peptide to the motor
domain to show that C-terminal peptide blocks MT binding. They then use a mutant (S715E)
in which the C-terminal peptide can't bind. This leads to a high affinity
interaction with the microtubule and decoration along the MTs length. The authors use
their data to support a model in which the C-terminal peptide reduces the affinity of
the MCAK motor domain for the lattice of the microtubule, allowing it to diffuse to the
ends where its activity is required.

Overall the data are better presented in this draft and the story is clearer. The
authors have added some nice experiments in which they remove the tubulin tails and tie
the data into their final model. Nevertheless, there are several issues that require
attention in order for the manuscript to be suitable for publication in
*eLife*.

Major experimental issues that need to be addressed:

There is still lingering concern about the role of the C-terminal domain in the
catalytic cycle of MCAK.

1) The authors show that the MCAK motor domain binds to microtubules with good binding
affinity and that the modified (Cys containing) C-terminal peptide does not affect the
binding affinity under reducing conditions. They have already clarified that the
Cys-containing C-terminal peptide does not bind MCAK under reducing conditions, i.e. it
does not trigger dimerization of MCAK. So the binding affinity for MTs they are
observing now is the binding affinity of the monomeric motor domain for MTs. Next, the
authors remove the reducing agent and observe that the fraction of MCAK bound to MTs has
decreased. They claim that this is the expected behavior and illustrate the result in
Figure 5 by showing that two equimolar
populations of MCAK form, one of which can bind MTs and one of which cannot. This
expectation is not well justified. How much of each of these two populations is
represented depends on the concentration of peptide and on the time the peptide is given
to form a covalent bond with MCAK, so that telling what is the expected result for this
experiment is very hard unless all the reaction parameters are at hand. If it is true
that the MCAK motor domain dimer:peptide has no or lower binding affinity for MT,
peptide addition should result in progressively lower MCAK binding as a function of the
concentration of C-terminal peptide and the time it is allowed to interact with
MCAK.

The problem is that this experiment is a “remnant” from the theory that
binding of the C-terminal tail to the motor domain is part of the catalytic cycle of
MCAK and from the authors' attempts to lock this putative intermediate. This
theory is downplayed in the present version and the authors could try to do the
experiment in a simpler and more convincing setting. They don't need a covalent
linker. They have created a full length MCAK with the S715E mutant, which they claim
prevents binding of the C-terminal peptide to the motor domain. They can simply measure
the binding affinity of this construct, which is because another segment of MCAK makes
it be dimeric (SEC-MALS analysis in the subsection headed “The C terminus
stabilizes full-length MCAK in solution”), in the presence of growing
concentrations of free C-terminal peptide. As there should be no intra-molecular
competition from the C-terminal tail (due to the S715E mutation) this experiment will
clarify the actual effects of the C-terminal peptide in the absence of other confounding
effects, particularly the underlying monomer-dimer equilibrium of the MCAK motor
domain.

2) Concerning Figure 5: The authors need to do a
control of MD no DTT, to show that the cross linker (rather than just lack of DTT) is
causing the effects.

Other issues:

3) The manuscript suffers from a number of minor mistakes which would need to be
corrected (listed below). It also could be improved by a clearer and more concise
writing style. Parts of it are really hard to understand without reading it multiple
times. In particular, the discussion of the experiments in Figure 5 is really cursory: the inferences are stated and the
reader is simply referred to the figure, for which the figure legend provides minimal
information. This is a key figure describing crucial experimental data, and the authors
should explain the experiments, the resulting data and inferences completely. They may
choose to break the figure up into multiple panels to achieve this end.

4) Figure 1: In the legend to Figure 1 the authors write: “The star
represents residual GST”. Only later in the manuscript did this become clear.
They need to explain that CT was made as a GST fusion and cleaved off. In Figure 1 and Figure 1 the order of the constructs is different. This is sloppy. The authors need
to correct it.

5) Figure 1–figure supplement 1: this figure looks suspicious. The intensity of
the peptide bands in the M+CT peak are very high relative to the CT bands. If
they are staining equally it suggests a ratio of 20 peptides to 1 motor domain. Also
there is a sharp drop off of intensity of the peptide band between fraction 10-11 and
also 15-16. This does not look like a gel filtration peak. The authors need to address
and possibly redo this gel. The fraction numbers on the gel need marking on the trace
(which currently shows only volume).

6) Figure 3: A number of structural elements are
referred to in the text that need to be shown on this figure: e.g. α1 and
β3 of the dimerization interface, L1 loop, the interaction between Glu172 and
Ala241/A is missing, Lys286/B is missing, structural elements of the ATPase site
(P-loop, switch I and II). It would also be helpful to add an additional overlay
structure to the figure that shows that these elements don't change upon C-tail
binding.

7) Figure 4: A quadruple mutant
(E711A/E712A/R716A/I718A) is shown in the figure but not discussed in the text. It
should be removed or talked about. If it is kept then the way it is written above the
gel needs to be improved as currently it is not clear that all 4 mutations were made in
the same construct.

8) Figure 5: Figure 5. The color lines in this figure are mislabelled. The magenta line
should be MD+CTsc no DTT (rather than MD). In the figure legend there is
reference to 6C, which should be 5D. The figure makes little sense otherwise. In this,
and other figures, the axes should be more clearly explained in the figure legend.
References in the text to the figures should also be readily understood by the reader.
What is “Bmax”, referred to in the text but not apparent in the
figure?

9) Figure 6: The cell images are too small and
bad quality. They need to be made brighter and larger in order to show what is
happening.

10) Figure 7: This figure is not currently
mentioned in the text and needs to be.

11) In the first paragraph of the Results section, the term “upstream” is
ambiguous. Consider using N or C terminal. The manuscript would be clearer if the
authors maintained a consistent way of referring to the parts of the model. Sometimes
they use CT and others “C-terminal domain” or “CT domain” or
“C terminus”.

12) The authors should spell out clearly how they prepared CT alone (with no GST) and
then clearly identify when they are using GST-CT and when CT.

13) In the second paragraph of the subsection “A conserved motif in the
C-terminal region of MCAK is essential for the C terminus-motor interaction”, the
term “head group” is confusing. The authors should state more clearly that
they are changing the size of the amino acid at position 715.

14) In the subsection entitled “The C terminus-motor domain interaction is
intramolecular”, the authors explain that removing the C-terminus of MCAK had no
effect in depolymerisation assays. They then state “It is possible that there is
a trade-off between microtubule binding affinity and tubulin removal at ends”.
This sentence needs to be expanded to make it clearer. As I understand it they think
there is no change in depolymerisation activity because of two counteracting effects
(increase in activity and decrease in accessibility to the MT ends due to too much
lattice binding). The authors could also consider moving this lack of change in activity
to the discussion. It made reading the paper difficult.

15) The whole section on the intramolecular interaction is very difficult to follow and
needs rephrasing to make it clearer. The authors may consider removing it as it does not
seem to add to the overall argument.

16) In the second paragraph of the Results section: It may be easier to discuss the
quantification of the binding affinity of the C-terminal peptide for the MCAK motor
domain after describing the observation that the peptide causes dimerization of the
C-terminal domain. The authors write: “…this measurement does not take
into account any existing equilibrium between the motor domains (Figure 2)”. In fact, the measurement results from the sum
of two distinct interactions, the dimerization of the MCAK motor domain and the binding
of the C-terminal peptide, an intrinsically cooperative interaction. It would be easier
to clarify this if the dimerization had been already introduced.

17) At the end of the subsection headed “One MCAK C terminus binds to two ATPase
domains”: “…we could not conclusively determine whether…
there were additional dimerization domains in the context of full length MCAK”.
The authors demonstrate later in the manuscript that MCAK forms dimers independently of
the C-terminal domain, so this sentence may be confusing.

18) In the subsection “The MCAK C terminus binds at the interface between two
motor domains”: There are four MCAK motor domains in the asymmetric unit of the
crystals, two of which we are told are bound as a near-symmetrical dimer to a single
peptide. The other two are not described, except for the explanation that they are not
bound to the peptide because of a crystal contact. Are the two additional subunits that
are not bound to the peptide in a dimeric arrangement? And also: If one considers only
the two MCAK subunits bound to the peptide, are the two motor subunits related by a
perfect (non-crystallographic) 2-fold axis? Clearly the presence of the peptide breaks
the symmetry, but limitedly to the motor domain, are they in a perfect dimer? If so, it
should be clearly stated.

19) At the end of the first paragraph of “The C terminus-motor domain interaction
is intramolecular”: “However it did raise the possibility that the CT
domain acts indirectly as an inhibitor and has an additional distinct cellular
function”. “It” is a series of negative results discussed in the
previous lines and showing that a point mutant that prevents binding of the CT has no
effect in vitro and in vivo on microtubule depolymerization. It seems to me that
emphasis here should be given to the negative results (some of which are not shown)
rather than on the authors' goal of proving the importance of their finding.

20) In the second paragraph of the subheading “The C terminus-motor domain
interaction is intramolecular”, the authors hypothesize that the reason why they
do not see a covalent dimer form under oxidizing conditions with full length MCAK E712C
is that the linkage is intra-molecular. It is possible, but the alternative explanation,
that the linkage is not formed at all in the context of the full-length protein, is more
plausible given the considerable number of residues lying in between the motor domain
and the C-terminus.

21) At the end of the subsection headed “A conserved motif in the C-terminal
region of MCAK is essential for the C terminus-motor interaction”:
“data” is plural.

22) In the beginning of the section entitled “The C terminus-motor domain
interaction is intramolecular”, the sentence starting with “It is possible
that there is a trade-off…” is obscure.

23) In the subsection headed “The C terminus stabilizes full-length MCAK in
solution”, it is not clear what is the evidence that MCAK S715S is less stable.
If this referred to the stability of the dimer, it would be useful that the authors
clarified how they reached this conclusion. As it stands, the inference that this mutant
is less stable should be removed, as it is not justified.

---

## [Author Response]

*Major experimental issues that need to be addressed*:

*There is still lingering concern about the role of the C-terminal domain in the
catalytic cycle of MCAK*.

*1) The authors show that the MCAK motor domain binds to microtubules with good
binding affinity and that the modified (Cys containing) C-terminal peptide does not
affect the binding affinity under reducing conditions. They have already clarified
that the Cys-containing C-terminal peptide does not bind MCAK under reducing
conditions, i.e. it does not trigger dimerization of MCAK. So the binding affinity
for MTs they are observing now is the binding affinity of the monomeric motor domain
for MTs. Next, the authors remove the reducing agent and observe that the fraction of
MCAK bound to MTs has decreased. They claim that this is the expected behavior and
illustrate the result in*
Figure 5
*by showing that two equimolar populations of MCAK form, one of which can bind
MTs and one of which cannot. This expectation is not well justified. How much of each
of these two populations is represented depends on the concentration of peptide and
on the time the peptide is given to form a covalent bond with MCAK, so that telling
what is the expected result for this experiment is very hard unless all the reaction
parameters are at hand. If it is true that the MCAK motor domain dimer:peptide has no
or lower binding affinity for MT, peptide addition should result in progressively
lower MCAK binding as a function of the concentration of C-terminal peptide and the
time it is allowed to interact with MCAK*.

*The problem is that this experiment is a “remnant” from the theory
that binding of the C-terminal tail to the motor domain is part of the catalytic
cycle of MCAK and from the authors' attempts to lock this putative
intermediate. This theory is downplayed in the present version and the authors could
try to do the experiment in a simpler and more convincing setting. They don't
need a covalent linker. They have created a full length MCAK with the S715E mutant,
which they claim prevents binding of the C-terminal peptide to the motor domain. They
can simply measure the binding affinity of this construct, which is because another
segment of MCAK makes it be dimeric (SEC-MALS analysis in the subsection headed
“The C terminus stabilizes full-length MCAK in solution”), in the
presence of growing concentrations of free C-terminal peptide. As there should be no
intra-molecular competition from the C-terminal tail (due to the S715E mutation) this
experiment will clarify the actual effects of the C-terminal peptide in the absence
of other confounding effects, particularly the underlying monomer-dimer equilibrium
of the MCAK motor domain*.

We have now performed this experiment. When we titrate increasing amounts of free CT
into the MCAK_S715E_-microtubule reaction, we gradually decreased the amount of
binding to the microtubule, revealing that the CT domain indeed reduces the affinity of
MCAK for microtubules.

*2) Concerning*
Figure 5*: The authors
need to do a control of MD no DTT, to show that the cross linker (rather than just
lack of DTT) is causing the effects*.

We have performed these experiments in Figure 6—figure supplement 3, where the motor domain (M) and the CT domain
(CT) were incubated in absence of DTT. We then performed cosedimentation and microtubule
depolymerization assays in presence of M+CT without reducing agent and we did not
observe any changes in MCAK activity. Therefore these data indicate that it is
specifically the covalent linkage between the motor and the CT_E712C_ in
absence of DTT that affects MCAK function.

*Other issues*:

*3) The manuscript suffers from a number of minor mistakes which would need to be
corrected (listed below). It also could be improved by a clearer and more concise
writing style. Parts of it are really hard to understand without reading it multiple
times. In particular, the discussion of the experiments in*
Figure 5
*is really cursory: the inferences are stated and the reader is simply referred
to the figure, for which the figure legend provides minimal information. This is a
key figure describing crucial experimental data, and the authors should explain the
experiments, the resulting data and inferences completely. They may choose to break
the figure up into multiple panels to achieve this end*.

We have now broken up Figure 5 into two
additional panels: one of them includes the experiment to major point 1, raised by the
reviewers. We have also modified the text and figure legends to explain the experiments
better.

*4)*
Figure 1*: In the legend
to*
Figure 1
*the authors write: “The star represents residual GST”. Only later
in the manuscript did this become clear. They need to explain that CT was made as a
GST fusion and cleaved off. In*
Figure 1
*and*
Figure 1
*the order of the constructs is different. This is sloppy. The authors need to
correct it*.

We have now stated that the tail domain (CT) is cleaved off from the GST and the GST is
removed (in the first paragraph of the Results and in the Methods section). We used a
smaller chemically synthesized peptide for complex crystallization and we have indicated
this in the text, that the peptide was chemically synthesized (please see the subsection
entitled “The MCAK C terminus binds at the interface between two motor
domains”). We have also changed the order of constructs in Figure 1.

*5) Figure 1–figure supplement 1: this figure looks suspicious. The
intensity of the peptide bands in the M+CT peak are very high relative to the
CT bands. If they are staining equally it suggests a ratio of 20 peptides to 1 motor
domain. Also there is a sharp drop off of intensity of the peptide band between
fraction 10-11 and also 15-16. This does not look like a gel filtration peak. The
authors need to address and possibly redo this gel. The fraction numbers on the gel
need marking on the trace (which currently shows only volume)*.

We apologize for this problem and we agree it was imperative to change the figure. We
have now repeated the gel filtration of the M, M+CT and
M+CT_mutants_ constructs on appropriate gels (Invitrogen 16% Tricine
gels) and have displayed the corresponding gels and elution profiles in Figures 1 and 4. We have removed the gel
filtration of the M+GST-CT constructs for clarity.

*6)*
Figure 3*: A number of
structural elements are referred to in the text that need to be shown on this figure:
e.g. α1 and β3 of the dimerization interface, L1 loop, the interaction
between Glu172 and Ala241/A is missing, Lys286/B is missing, structural elements of
the ATPase site (P-loop, switch I and II). It would also be helpful to add an
additional overlay structure to the figure that shows that these elements
don't change upon C-tail binding*.

We thank the reviewers for these excellent suggestions. They really help to understand
the functional aspects of our MCAK structure better. We have now included new figures
(Figure 3—figure supplement 1) with
overlays of the mouse MCAK structure published by the Hirokawa group in 2004, with our
MCAK structure. We then show that the ATPase site and the switch I and II regions are
unperturbed while the neck linker region changes orientation around Lys258, with His257
interacting with the C-terminus. We have also included annotations to α1 and
β3 in the dimerization interface and generated a figure that highlights the
interaction between Lys286/B and Glu244/A, as it is only present when the CT domain is
bound to the motor in our structure.

*7)*
Figure 4*: A quadruple
mutant (E711A/E712A/R716A/I718A) is shown in the figure but not discussed in the
text. It should be removed or talked about. If it is kept then the way it is written
above the gel needs to be improved as currently it is not clear that all 4 mutations
were made in the same construct*.

We have now removed this from the figure and repeated the pulldown assay in Figure 4. At the same time, we have included the
experiment in Figure 4–figure supplement 1A showing that the CT_S715A_
binds to the motor into Figure 4.

*8)*
Figure 5*:*
Figure 5*. The color
lines in this figure are mislabelled. The magenta line should be MD+CTsc no
DTT (rather than MD). In the figure legend there is reference to 6C, which should be
5D. The figure makes little sense otherwise. In this, and other figures, the axes
should be more clearly explained in the figure legend. References in the text to the
figures should also be readily understood by the reader. What is
“Bmax”, referred to in the text but not apparent in the
figure*?

We thank the reviewers and have now corrected this mislabeled figure. We have also
relabeled the graph axes to explain more clearly what they represent and removed the
reference to “Bmax” in the text. We have also edited the references to the
figures.

*9)*
Figure 6*: The cell
images are too small and bad quality. They need to be made brighter and larger in
order to show what is happening*.

We have now improved the presentation of the images to show the EB3 and MCAK staining
more closely. We have also added linescan profile averages for the EB3 and MCAK
localization on microtubule plus ends.

*10)*
Figure 7*: This figure is
not currently mentioned in the text and needs to be*.

We apologize for this mistake and have now mentioned Figure 7 (now Figure 8) in the
text.

*11) In the first paragraph of the Results section, the term
“upstream” is ambiguous. Consider using N or C terminal. The manuscript
would be clearer if the authors maintained a consistent way of referring to the parts
of the model. Sometimes they use CT and others “C-terminal domain” or
“CT domain” or “C terminus”*.

We have now removed the term “upstream”. Throughout the text, we have more
consistently labeled the C-terminus of MCAK construct as the CT domain. We apologize for
this inconsistency.

*12) The authors should spell out clearly how they prepared CT alone (with no
GST) and then clearly identify when they are using GST-CT and when CT*.

We have now described in the Methods section more clearly how the CT domain is prepared
and how the GST is removed. We have also carefully annotated the figures and have tried
to avoid using GST-CT when possible.

*13) In the second paragraph of the subsection headed “A conserved motif
in the C-terminal region of MCAK is essential for the C terminus-motor
interaction”, the term “head group” is confusing. The authors
should state more clearly that they are changing the size of the amino acid at
position 715*.

We have now changed the term “head group” to “side
chain”.

*14) In the subsection headed “The C terminus-motor domain interaction is
intramolecular”, the authors explain that removing the C-terminus of MCAK had
no effect in depolymerisation assays. They then state “It is possible that
there is a trade-off between microtubule binding affinity and tubulin removal at
ends”. This sentence needs to be expanded to make it clearer. As I understand
it they think there is no change in depolymerisation activity because of two
counteracting effects (increase in activity and decrease in accessibility to the MT
ends due to too much lattice binding). The authors could also consider moving this
lack of change in activity to the discussion. It made reading the paper
difficult*.

We have now changed this sentence to: “However there are limitations to this
assay, in which we can only measure rate of microtubule depolymerization. It is possible
that a change in MCAK microtubule binding affinity will have a counteracting effect on
MCAK diffusion rate or the rate of tubulin removal at ends and that as a consequence.
The overall depolymerase activity that our assay measures is unchanged, as previously
shown (9).”

*15) The whole section on the intramolecular interaction is very difficult to
follow and needs rephrasing to make it clearer. The authors may consider removing it
as it does not seem to add to the overall argument*.

We have now removed this section to streamline the paper.

*16) In the second paragraph of the Results section: It may be easier to discuss
the quantification of the binding affinity of the C-terminal peptide for the MCAK
motor domain after describing the observation that the peptide causes dimerization of
the C-terminal domain. The authors write: “…this measurement does not
take into account any existing equilibrium between the motor domains (*Figure 2*)”. In
fact, the measurement results from the sum of two distinct interactions, the
dimerization of the MCAK motor domain and the binding of the C-terminal peptide, an
intrinsically cooperative interaction. It would be easier to clarify this if the
dimerization had been already introduced*.

We apologize for discussing motor dimerization before we introduced the data relative to
it in the paper and we thank the reviewers for this comment. We have now moved the
section quantifying the affinity of the peptide for the motor so that it follows the
section describing that the CT domain induces motor dimerization. We have clarified that
the affinity measured represents the sum of two distinct affinities, of which CT domain
binding is cooperative.

*17) At the end of the subsection headed “One MCAK C terminus binds to two
ATPase domains”: “…we could not conclusively determine
whether… there were additional dimerization domains in the context of full
length MCAK”. The authors demonstrate later in the manuscript that MCAK forms
dimers independently of the C-terminal domain, so this sentence may be
confusing*.

We have now included “from the above experiments” to highlight that the
data described in that paragraph does not indicate whether MCAK is still dimeric or
not.

*18) In the subsection “The MCAK C terminus binds at the interface between
two motor domains”: There are four MCAK motor domains in the asymmetric unit
of the crystals, two of which we are told are bound as a near-symmetrical dimer to a
single peptide. The other two are not described, except for the explanation that they
are not bound to the peptide because of a crystal contact. Are the two additional
subunits that are not bound to the peptide in a dimeric arrangement? And also: If one
considers only the two MCAK subunits bound to the peptide, are the two motor subunits
related by a perfect (non-crystallographic) 2-fold axis? Clearly the presence of the
peptide breaks the symmetry, but limitedly to the motor domain, are they in a perfect
dimer? If so, it should be clearly stated*.

We thank the reviewers for this excellent point. We have now changed the text to reflect
the structure better and have included this information and an additional figure of the
CT binding site on the dimer that is not occupied by the CT domain.

*19) At the end of the first paragraph of “The C terminus-motor domain
interaction is intramolecular”: “However it did raise the possibility
that the CT domain acts indirectly as an inhibitor and has an additional distinct
cellular function”. “It” is a series of negative results
discussed in the previous lines and showing that a point mutant that prevents binding
of the CT has no effect in vitro and in vivo on microtubule depolymerization. It
seems to me that emphasis here should be given to the negative results (some of which
are not shown) rather than on the authors' goal of proving the importance of
their finding*.

We have measured the activity of MCAK and MCAK mutant such as MCAK_S175E_ using
an assay first described by [30]. The assay measures microtubule depolymerase activity in cells, by
quantifying the microtubule fluorescence. However, while expression of MCAK
depolymerizes microtubules, overexpression leads to microtubule bundling and
abnormalities. We believe there are some limitations to this assay. Thus we could not
measure with confidence significant differences in MCAK microtubule depolymerase
activity upon removal of the CT domain and publish this data. We also used the
PlusTipTracker (Danuser group) and a cell line expressing 2xGFP-EB3 to detect
MCAK-dependent changes in microtubule-dynamics, by transiently transfecting MCAK and
MCAK_S715E_. However, the sample heterogeneity was in the order of magnitude
of the changes recorded for the distinct mutants and transfection of GFP-MCAK rescued
growth lifetime, but not growth speed. Therefore we did not include this data.

Finally, another paper (23)
reported that removal of the CT domain resulted in a decrease in microtubule
depolymerase activity, thus the effect of the CT domain on MCAK activity remains under
debate. We have now included this information and the reference to the paper in this
section.

*20) In the second paragraph of the subheading “The C terminus-motor
domain interaction is intramolecular”*, *the authors
hypothesize that the reason why they do not see a covalent dimer form under oxidizing
conditions with full length MCAK E712C is that the linkage is intra-molecular. It is
possible, but the alternative explanation, that the linkage is not formed at all in
the context of the full-length protein, is more plausible given the considerable
number of residues lying in between the motor domain and the C-terminus*.

We believe that we do form the crosslink because we see a reduction in the affinity of
MCAK_E712C_ for microtubules in absence of DTT (Figure 5–figure
supplement 1D). However, Cys287 (Cys283 in mouse) has also been reported to make a
disulfide bond with Cys245 (Cys241 in mouse) in the murine MCAK crystal structure. Thus
it is possible, that in the context of the full length MCAK and in absence of DTT, we
generate at least two MCAK conformations of neck-motor complexes that make the analysis
of this MCAK_E712C_ mutant more complex.

*21) At the end of the subsection headed “A conserved motif in the
C-terminal region of MCAK is essential for the C terminus-motor interaction”:
“data” is plural*.

We apologize and have now corrected this grammatical mistake.

*22) In the beginning of the section entitled “The C terminus-motor domain
interaction is intramolecular”*, *the sentence starting with
“It is possible that there is a trade-off…” is
obscure*.

We have now clarified this sentence in the text.

*23) In the subsection headed “The C terminus stabilizes full-length MCAK
in solution”, it is not clear what is the evidence that MCAK S715S is less
stable. If this referred to the stability of the dimer, it would be useful that the
authors clarified how they reached this conclusion. As it stands, the inference that
this mutant is less stable should be removed, as it is not justified*.

The recovery rate from SEC-MALS was 33%, which implies some of the protein was not
stable, but we did not include that data and we apologize for it. However, we have now
removed the SEC-MALS data for MCAK_S715E_ and only included the gel filtration
elution profile of MCAK_S715E._